# Sexual coordination in a whole-brain map of prairie vole pair bonding

Morgan L Gustison[1,2], Rodrigo Muñoz-Castañeda[3,4], Pavel Osten[3]*, Steven M Phelps[1,5]*

[1]Department of Integrative Biology, The University of Texas at Austin, Austin, United States; [2]Department of Psychology, Western University, London, Canada; [3]Cold Spring Harbor Laboratory, Cold Spring Harbor, United States; [4]Appel Alzheimer's Disease Research Institute, Feil Family Brain and Mind Research Institute, Weill Cornell Medicine, New York, United States; [5]Institute for Neuroscience, The University of Texas at Austin, Austin, United States

**\*For correspondence:**
pavel.osten@gmail.com (PO);
sphelps@utexas.edu (SMP)

**Abstract** Sexual bonds are central to the social lives of many species, including humans, and monogamous prairie voles have become the predominant model for investigating such attachments. We developed an automated whole-brain mapping pipeline to identify brain circuits underlying pair-bonding behavior. We identified bonding-related c-Fos induction in 68 brain regions clustered in seven major brain-wide neuronal circuits. These circuits include known regulators of bonding, such as the bed nucleus of the stria terminalis, paraventricular hypothalamus, ventral pallidum, and prefrontal cortex. They also include brain regions previously unknown to shape bonding, such as ventromedial hypothalamus, medial preoptic area, and the medial amygdala, but that play essential roles in bonding-relevant processes, such as sexual behavior, social reward, and territorial aggression. Contrary to some hypotheses, we found that circuits active during mating and bonding were largely sexually monomorphic. Moreover, c-Fos induction across regions was strikingly consistent between members of a pair, with activity best predicted by rates of ejaculation. A novel cluster of regions centered in the amygdala remained coordinated after bonds had formed, suggesting novel substrates for bond maintenance. Our tools and results provide an unprecedented resource for elucidating the networks that translate sexual experience into an enduring bond.

## eLife assessment

This is an **important** study using 3D mapping of neuronal activation throughout the brain after pair-bonding in the monogamous vole, which can be broadly applied to other species and behaviors. The authors provide **compelling** evidence that there is some synchrony between male and female partners that have formed a pair bond, the strength of which is based on the number of ejaculations received by the female. Same-sex pairs also form a pair bond and were found to have activation in the same brain regions as mixed sex couples. An overall low level of sex differences in the degree and location of brain activation was observed, which was unexpected. This work will be of interest to those interested in social behavior and its neural mechanisms, or brain systems or behavior more broadly.

## Introduction

Bonds are essential to the social lives of many species, enabling individuals to coordinate parental care, territorial defense, or other shared activities (*Emlen and Oring, 1977*; *Marlowe, 2000*; *Trivers, 1972*). Among humans, friendships, and a happy marriage protect against a wide range of stressors

and their sequelae (*Baumeister and Leary, 1995*; *Hawkley and Cacioppo, 2010*), while social integration is one of the strongest predictors of reduced morbidity and mortality risk (*Snyder-Mackler et al., 2020*). Recent evidence suggests social buffering against stress-related disease can be found in many other mammals as well, including species of nonhuman primates, rodents, ungulates, and hyrax (*Snyder-Mackler et al., 2020*). Nonhuman animals offer unique insights into the mechanisms of social attachment and their consequences.

The neurobiology of bonding is most studied in the prairie vole, a socially monogamous rodent in which females and males form bonds, share a nest, and raise young together (*Lieberwirth and Wang, 2014*; *Young and Wang, 2004*). In prairie voles, as in many pair-bonding species, bonds form in response to courtship and repeated mating (*Carter et al., 1995*; *Getz et al., 1993*); indeed, copulation itself is regarded as a form of courtship that allows mutual assessment and coordinates reproduction (*Dewsbury, 1988*; *Eberhard, 1996*). Because of these characteristics, prairie voles have become an excellent model species to study neural circuits that underpin affiliative behaviors (*Kenkel et al., 2021*). Classic studies emphasized the roles of neuropeptides in reward circuitry (*Aragona et al., 2006*; *Young and Wang, 2004*), work that has been refined to reveal that neural ensembles in the nucleus accumbens encode approach behavior between mates (*Scribner et al., 2020*), and the strength of functional connections between nucleus accumbens and prefrontal cortex predicts social contact (*Amadei et al., 2017*). Mated prairie voles exhibit empathy-like consolation behavior that reduces markers of distress in both members of a pair (*Burkett et al., 2016*). In one common neuro-anatomical model, some 18 different brain regions shape aspects of memory, reward, and approach that are essential to bond formation and its consequences (*Walum and Young, 2018*). Despite this extraordinary work, we still have an incomplete understanding of the circuits that enable the experience of mating to become an enduring bond. Building upon this foundation will require a shift from a piece-meal to systems-wide perspective.

In several species, including wasps, fish, bats, mice, and humans, researchers using a variety of biological measures find that social interactions are accompanied by coordinated neural states (*Hasson et al., 2012*; *Kingsbury et al., 2019*; *Kinreich et al., 2017*; *Long et al., 2020*; *Vu et al., 2020*; *Zhang and Yartsev, 2019*). In fighting fish (*Betta splendens*) and paper wasps (*Polistes fuscatus*), shared brain-transcriptomic signatures occur in individuals immediately after competitive interactions (*Uy et al., 2021*; *Vu et al., 2020*). In Egyptian fruit bats (*Rousettus aegyptiacus*) and lab mice, freely interacting individuals have correlated neural activity in the frontal cortex, and increases in inter-brain correlations predict whether subsequent interactions occur (*Kingsbury et al., 2019*; *Zhang and Yartsev, 2019*). In humans, inter-brain EEG recordings become synchronized in a variety of interactive contexts (*Hasson et al., 2012*), such as during social gaze and positive affect in couples (*Kinreich et al., 2017*). Such data have led to the inference that closely interacting dyads, including bonded individuals, coordinate aspects of their physiological and neural states (*Long et al., 2020*).

In contrast to the emphasis on shared brain states, sex differences in neuroendocrine mechanisms mean that not only are behaviors often sexually dimorphic, but even if the sexes exhibit similar behaviors, a baseline difference in brain function may require distinct behavioral mechanisms (*De Vries, 2004*). In the monogamous oldfield mouse, *Peromyscus polionotus*, for example, genome-wide association studies implicate different mechanisms in the regulation of female and male parental care (*Bendesky et al., 2017*). Similarly, the 'dual function hypothesis' has suggested that dimorphic mechanisms may underlie bonding or parental behaviors more generally (*De Vries, 2004*). To understand attachment, it is essential to examine neural function across the entire brain (*López-Gutiérrez et al., 2021*; *Yee et al., 2016*), and to explore individual and sex differences in such circuits as bonds form.

To systematically examine the mechanisms of pair-bond formation, we developed a whole-brain imaging and computational analysis pipeline that includes the first 3D histological atlas of the prairie vole brain. This atlas and analysis pipeline enables the high-throughput, automated counting of cell markers throughout the prairie vole brain. We used this tool to test for sexual dimorphism in the structure of prairie vole brains, and to compare the gross anatomy of the prairie vole brain to the laboratory mouse. In order to map regions and circuits active during bonding, we next quantified immediate early gene (IEG) induction in 824 brain regions at key times in bond formation. This represents the first unbiased identification of circuits active during bonding. Doing so allowed us to examine how the experience of mating finds its way into bonding circuits. It also allowed us to rigorously test two alternative



**Video 1.** Prairie vole reference brain. The prairie vole reference brain from light-sheet fluorescence microscopy (LSFM) imaging is shown for coronal cross-sections (rostral to caudal) and in a 3D view. Then, coronal cross-sections are shown for the prairie vole atlas (left) alongside whole-brain staining for somatostatin (SST, middle) and fluorescent Nissl (NeuroTrace, right).

https://elifesciences.org/articles/87029/figures#video1

hypotheses: that mating and bonding promote coordinated changes in the brains of mated pairs; and conversely, that sexually dimorphic circuits underlie bonding in females and males.

## Results

### A novel whole-brain imaging pipeline for the prairie vole

We began by generating a common coordinate framework (CCF) for the prairie vole brain by iteratively averaging tissue autofluorescence from 190 brains imaged using light-sheet fluorescence microscopy (LSFM) (*Video 1*). Each of the brains was co-registered into this coordinate framework for computational analyses. Next, we registered an LSFM-based CCF of the mouse brain onto the prairie vole CCF, enabling us to apply anatomical labels of the Allen Reference Atlas (ARA; *Figure 1A*, *Figure 1—figure supplement 1*, and *Video 1*; *Dong, 2008*). This alignment revealed that the prairie vole brain (*n* = 190, 94 males and 96 females) is ~30% larger than the mouse brain (*n* = 108 males, negative binomial test: *z* = 36.957, false discovery rate [FDR] corrected *q* < 0.0001) and distinct in shape, but the relative volumes of brain regions were consistent across species (*q* > 0.05 for all regions) and showed no evidence of sexual dimorphism (*q* > 0.05 for all regions) (*Figure 1A*, *Figure 1—figure supplement 1*, *Supplementary file 1*, and *Supplementary file 2*).

To validate and refine the Allen Reference Atlas (ARA) anatomical borders in the prairie vole brain, we performed whole-brain Nissl staining as well as iDISCO immunolabeling targeting the cell-specific markers tyrosine hydroxylase, parvalbumin, and somatostatin; we then aligned these images onto the prairie vole CCF (*Figure 1B, C* and *Figure 1—figure supplement 1*). Lastly, we adapted for the prairie vole our computational pipeline for whole-brain detection and statistical comparisons of c-Fos+ neurons in LSFM-imaged brains (*Kim et al., 2015*; *Kim et al., 2016*; *Renier et al., 2016*; *Figure 1C*). With a robust atlas and analysis tools in place, we conducted a detailed study of the time course of mating and bond formation to examine IEG induction across the prairie vole brain and test our hypotheses.

### Pair bonding involves a dynamic repertoire of social interactions

Sexually responsive voles typically mate within the first hour of pairing, and repeated mating over ~6 hr initiates partner preferences characteristic of attachment (*DeVries and Carter, 1999*). The bond stabilizes between ~12 and 24 hr, leading to prolonged changes in attachment and related behaviors (*DeVries and Carter, 1999*; *Williams et al., 1992*). With these milestones in mind, we precisely manipulated mating experience and examined how repeated sexual behaviors lead to a bond (*Figure 2A*, *Figure 2—figure supplement 1*, *Figure 2—figure supplement 2*).

To coordinate the timing of mating, all subjects were isolated for 4–5 days, females were brought into estrus, and both sexes were screened for sexual receptivity (see *Figure 2A*). Subjects assigned to the bonding condition were paired with a novel opposite-sexed individual; to control for non-sexual social affiliation, remaining subjects were re-paired with a same-sex sibling who had been a cagemate prior to isolation. Members of each pair were isolated on either side of a divider for 2 hr. Following this acclimation, the divider was removed, and the pair could interact freely (*Figure 2A*). Behavior sessions were terminated following a fixed timeline: animals were euthanized just before barrier removal (0 hr), following initial mating (2.5 hr), after initial bond formation (6 hr), or after bond stabilization (22 hr). Automated behavioral measures, such as proximity, vocalization, and relative movement were scored throughout the behavioral sessions. We performed detailed manual scoring for 1 hr focal intervals beginning 2 hr before euthanasia – a timeframe that reveals the behavioral states of animals during

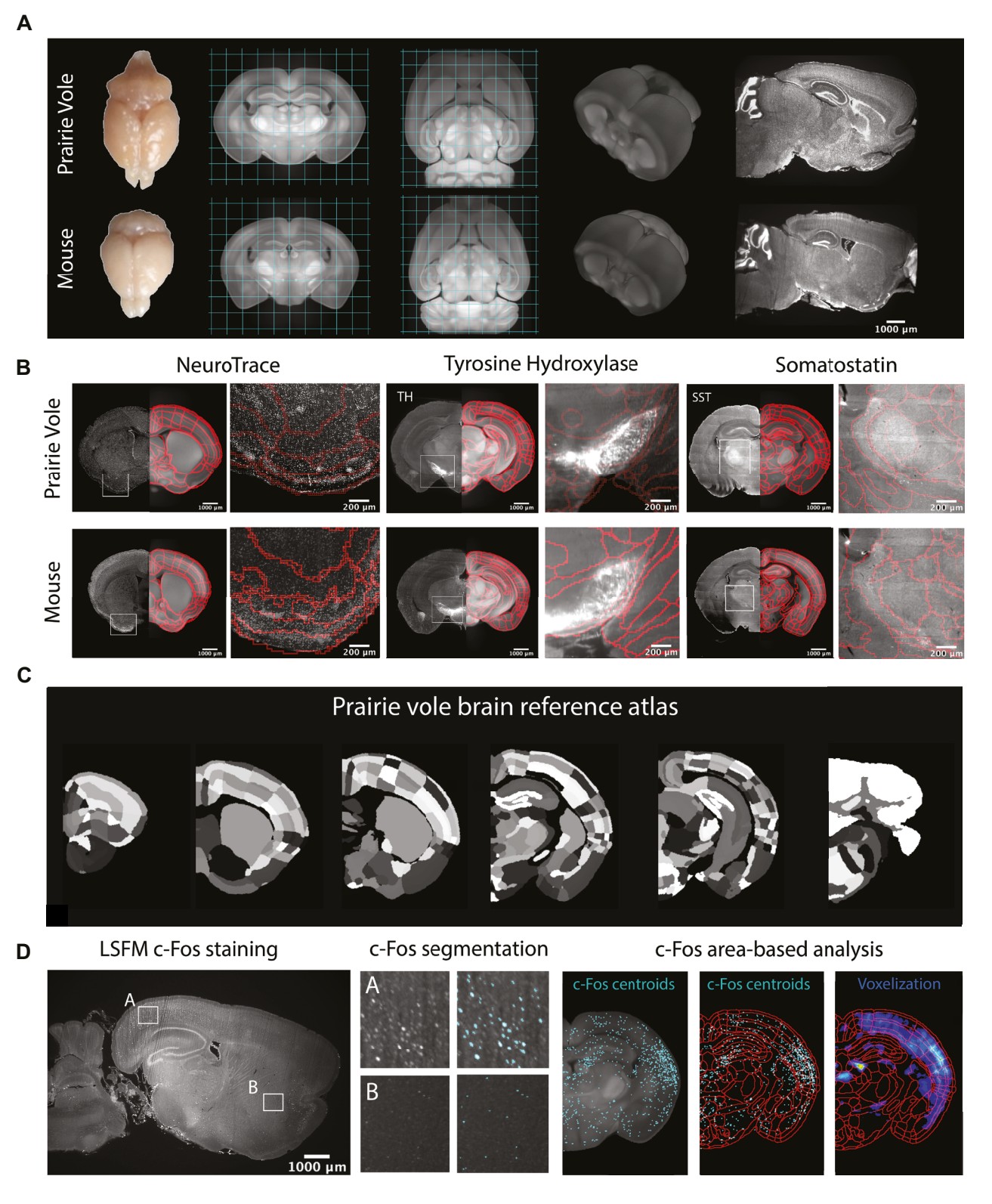

**Figure 1.** Prairie vole reference brain and atlas generation for automatic c-Fos analysis. (**A**) Generation of prairie vole (upper row) and mouse (lower row) reference brains was done with light-sheet fluorescence microscopy (LSFM) imaging. Shown is the top view of both prairie vole and mouse brains after perfusion. Also shown are cross-sections of coronal and horizontal views of prairie vole and mouse reference brains built after the co-registration of nearly 200 brains per species (approximately 190 voles and 150 mice). The prairie vole whole brain was 1.39 times larger than the mouse brain

*Figure 1 continued on next page*

*Figure 1 continued*

(see *Figure 1—figure supplement 1* and *Supplementary file 1*). Both prairie vole and mouse brains underwent 3D renderization. Also shown are sagittal views of prairie vole and mouse whole-brain fluorescent Nissl (NeuroTrace) staining imaged with LSFM. The scale bar represents a distance of 1,000 μm. (**B**) Whole-brain staining of both prairie vole (upper row) and mouse (lower row) brains were registered to the reference brain. In red, boundaries of registered mouse reference atlas are plotted onto both prairie vole and mouse reference brains. NeuroTrace, tyrosine hydroxylase (TH), and somatostatin (SST) were registered and overlaid onto the atlases for validation. (**C**) Coronal sections of the resulting prairie vole atlas after manual validation. Scale bars represent distances of 1,000 μm and 200 μm for whole section and close-up views, respectively. (**D**) Overview of LSFM prairie vole c-Fos analysis pipeline. Shown in the left panel is a sagittal section of prairie vole c-Fos immunolabeling imaged with LSFM (scale bar represents a distance of 1,000 μm). Shown in the center panel are detailed views of two brain locations immunolabeled with c-Fos and overlaid with the resulting segmentation. Shown in the right panel is an area-based analysis of c-Fos+ cells. All c-Fos+ cell centroids are registered to the prairie vole reference brain and analyzed using the new prairie vole reference atlas. For each brain, a voxel representation is generated of all c-Fos+ cells in the same prairie vole reference space and overlaid with the reference atlas.

The online version of this article includes the following figure supplement(s) for figure 1:

**Figure supplement 1.** Validation of the prairie vole reference atlas.

IEG induction (*Figure 2B, C*, *Figure 2—figure supplement 1*, *Figure 2—figure supplement 2*, *Figure 2—figure supplement 3*, and *Supplementary file 3*).

Mate and sibling pairs did not differ in locomotor activity or time spent in side-by-side contact ('huddling'; two sample *t*-tests: velocity, $n = 94$ mates and 96 siblings, $t = 1.592$, false discovery rate [FDR]-corrected $q = 0.170$, huddling, $n = 47$ mate and 48 sibling dyads, $t = 1.122$, $q = 0.3663$; *Figure 2D*). Mate pairs progressed through known stages of mating behavior, showing elevated rates of anogenital investigation and vocalization (*t*-tests: investigation, $n = 94$ mates and 96 siblings, $t = 4.184$, $q = 0.0003$, vocalizations, $n = 48$ mate and 48 sibling dyads, $t = 4.708$, $q = 0.0003$); males moved more often toward females, while females moved away from their partners (paired *t*-test: net movement, $n = 47$ females and 47 males, $t = 3.127$, $q = 0.0138$), a behavior consistent with female copulatory pacing (*Pfaus et al., 2001*). Consummatory aspects of mating – mounting, intromission and ejaculation – were common in both the 2.5 and 6 hr focal windows; by 22 hr mating was rare, and levels of male–female huddling resembled those of same-sex siblings housed together since birth (*Figure 2E*). This profile is consistent with literature on the timing and process of prairie vole bond formation (*DeVries and Carter, 1999*; *Williams et al., 1992*).

## A brain-wide functional neural network for pair bonding reveals seven major neuroanatomical clusters, with special prominence for regions of the bed nucleus of the stria terminalis and hypothalamus

Next we measured brain-wide c-Fos immunostaining, a common proxy for neuronal activity and plasticity (*Sheng and Greenberg, 1990*). To identify brain areas that differed between mate-paired and sibling controls, we used a generalized linear model (GLM) to compare two alternative models. A null model included terms for sex, time, and experiment block; a full model included these variables, as well as pairing status and two interaction terms (pairing × sex, pairing × time). Brain-wide comparisons using voxels or regions of interest (ROIs) were largely concordant (*Figure 3A, B*, *Figure 3—figure supplement 1*, and *Video 2*), revealing an extensive but specific network of brain regions active during mating and bonding. The brain atlas is organized hierarchically, and of 99 ROIs that differed significantly (permutation test: $n = 189$ animals, permutations = 10,000, FDR-corrected $q < 0.1$; *Figure 3B* and *Supplementary file 4*), 68 regions were anatomically distinct.

A hierarchical cluster analysis assigned these 68 ROIs into 8 groups (*Figure 3C, D*, *Figure 3—figure supplement 2*, and *Figure 3—figure supplement 3*). Each of these groups, or 'clusters', included multiple brain regions but often centered on a specific structure or group of structures. The purple 'BST' (bed nucleus of the stria terminalis) cluster contained regions of the preoptic area and periventricular nucleus of the hypothalamus, as well as multiple substructures within the BST. The blue 'POA-VMH' (preoptic area, ventromedial hypothalamus) and light blue 'PVH' (paraventricular hypothalamus) clusters contained multiple regions within the hypothalamus, including the paraventricular nucleus of the hypothalamus, as well as regions of the preoptic area. The green 'PFC' (prefrontal cortex) cluster was composed of regions within both the prefrontal cortex and olfactory areas. The orange 'Amyg' (amygdala) cluster contained the lateral hypothalamus and various olfactory related regions, but the majority of the cluster consisted of amygdalar nuclei such as cortical and medial amygdala and the

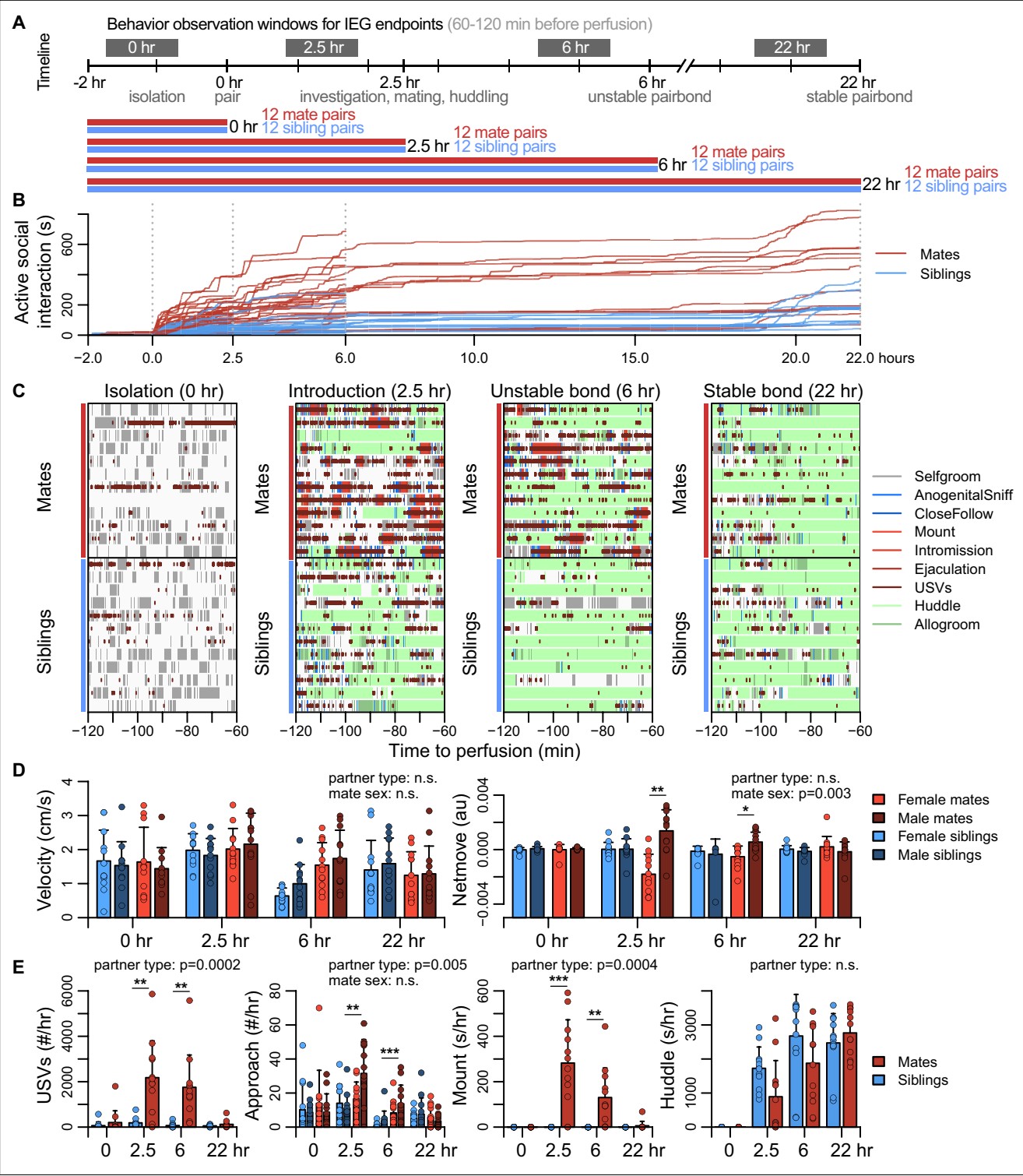

**Figure 2.** Study design and development of the prairie vole pair bond. (**A**) Schematic of the experiment design is shown, where social behaviors (during 1 hr observations, black blocks) and immediate early gene (IEG) expression patterns are compared between mate pairs and siblings across time. (**B**) Continuous automated tracking is shown for active social interactions (i.e., an index of social investigation and mating) in mate pairs (red lines) and sibling pairs (blue lines). (**C**) Time courses are shown for specific social behaviors during 1 hr observation windows that correspond to IEG expression, with each row representing one pair. Social behaviors are overlaid onto self-grooming (grey), and ultrasonic vocalizations (USVs, short dark red ticks) are overlaid onto all behaviors. Other behaviors shown include anogenital investigation (light blue), close follows (dark blue), mount (very light red), intromissions (light red), ejaculation (red), huddle (light green), and allogroom (dark green). (**D**) Plots show group differences (mean ± standard deviation

*Figure 2 continued on next page*

*Figure 2 continued*

[SD]) in individual activity level (velocity) and movement relative to the partner (net move, positive values indicate movement toward the partner). (**E**) Group differences (mean ± SD) are shown for vocal behavior (ultrasonic vocalizations [USVs]), proximity seeking (approach), mating (mount), and side-by-side contact (huddle). For (**D, E**), mate pairs are in red and sibling pairs in blue for dyadic-level behaviors (USVs, mount, and huddle). Females are a lighter hue and males a darker hue for behaviors measured on an individual level (velocity, netmove, approach). *T*-tests were used to compare mate animals (*n* = 94) or mate dyads (USVs: *n* = 48; mount and huddle: *n* = 47) with sibling animals (*n* = 96) or sibling dyads (*n* = 48). Paired *t*-tests were used to compare female mates (*n* = 47) with male mates (*n* = 47). For behaviors with group-level effects, follow-up t-tests were performed for each timepoint (*n* = 11-12 animals/group or *n* = 22-24 dyads/group; significance represented by * p<0.05, ** p<0.01, *** p<0.001). Descriptions of vole behaviors are listed in **Supplementary file 3**.

The online version of this article includes the following figure supplement(s) for figure 2:

**Figure supplement 1.** Automated tracking of social behavior states.

**Figure supplement 2.** Time course of social behaviors during pairing.

**Figure supplement 3.** Associations between behavioral states and types of social interaction.

anterior amygdala area. The light orange 'AUD' (auditory cortex) cluster contained some regions within auditory cortex, and the red 'Thal' (thalamus) cluster involved a variety of thalamic regions. The light purple 'AOB' (accessory olfactory bulb) cluster contained minimally correlated regions (nucleus y, accessory facial motor nucleus, and AOB glomerular layer), and so we do not consider this group to be one of the major clusters. Comparing the regions with connectivity reported in the ARA mouse 'connectome' (**Knox et al., 2019**) suggests that six major clusters in our dataset (BST, POA-VMH, PVH, PFC, Amyg, and AUD) are anatomically connected (permutation test with 10,000 shuffled datasets, p = 0.0001, **Figure 3—figure supplement 4**).

Although the nucleus accumbens did not survive multiple test corrections in our ROI analysis (*n* = 189 animals, F = 2.936, FDR *q* = 0.1747), it was significant in the univariate analysis (p = 0.0304), particularly when focused on the 2.5 and 6 hr timepoints (two sample *t*-test: *n* = 47 mates and 47 siblings, *t* = 2.530, p = 0.0138, **Video 2**). Furthermore, voxel-level comparisons revealed significant sites within the ventral striatum and the posterior nucleus accumbens (**Figure 2A**, **Figure 3—figure supplement 1**, and **Video 2**).

## Major dimensions of neural and behavioral variation are coordinated among mated pairs

Canonical correlation analysis (CCA) reveals the latent correlational structure within two sets of variables, and so is well suited to compare the principal dimensions of behavioral variation to its neural counterparts (**Figure 4**, **Figure 4—figure supplement 1**, and **Figure 4—figure supplement 2**). We found that the first canonical correlate (CC1), which defines the largest axis of shared variation in brain and behavior, loaded highly on the BST, POA-VMH, and PVH clusters, as well as on mating-related behaviors (**Figure 4A–D**). CC1 scores captured responses to mating and bonding in the 2.5 and 6 hr timepoints (**Figure 4A, B**). The second canonical correlate (CC2) captured differences between animals who were isolated or paired and loaded particularly high on the limbic cortical cluster (PFC; **Figure 4B**, **Figure 4—figure supplement 1**).

The first two CC dimensions showed no evidence of sexual dimorphism, while the third dimension suggested subtle but non-significant differences between male and female mates (**Figure 4—figure supplement 1**). A formal two-model comparison isolating the effect of the sex × pairing interaction revealed no regions that differed following FDR correction (permutation tests: *q* > 0.1 for all tests). If we limit the comparison to just those 68 regions identified as responsive to pairing, we find 29 unique regions that exhibit evidence of a sex × pairing interaction, although these effects are much weaker than those identified in our above analysis of pairing (**Video 2**).

Although differences between females and males were modest, the CCA suggested substantial variation across mated pairs. We used these data to test whether pairing involved a coordinated change in neural activation across brain regions. We find that during mating and bond formation (2.5–6 hr), activity is strongly correlated across regions (**Figure 4E, F** and **Figure 4—figure supplement 3**). Similar to these patterns distributed across putative pair-bonding regions, CC1 brain scores show a strong correlation between female and male mates (Pearson correlation: df = 22, *r* = 0.93, p < 0.0001, **Figure 4G**).

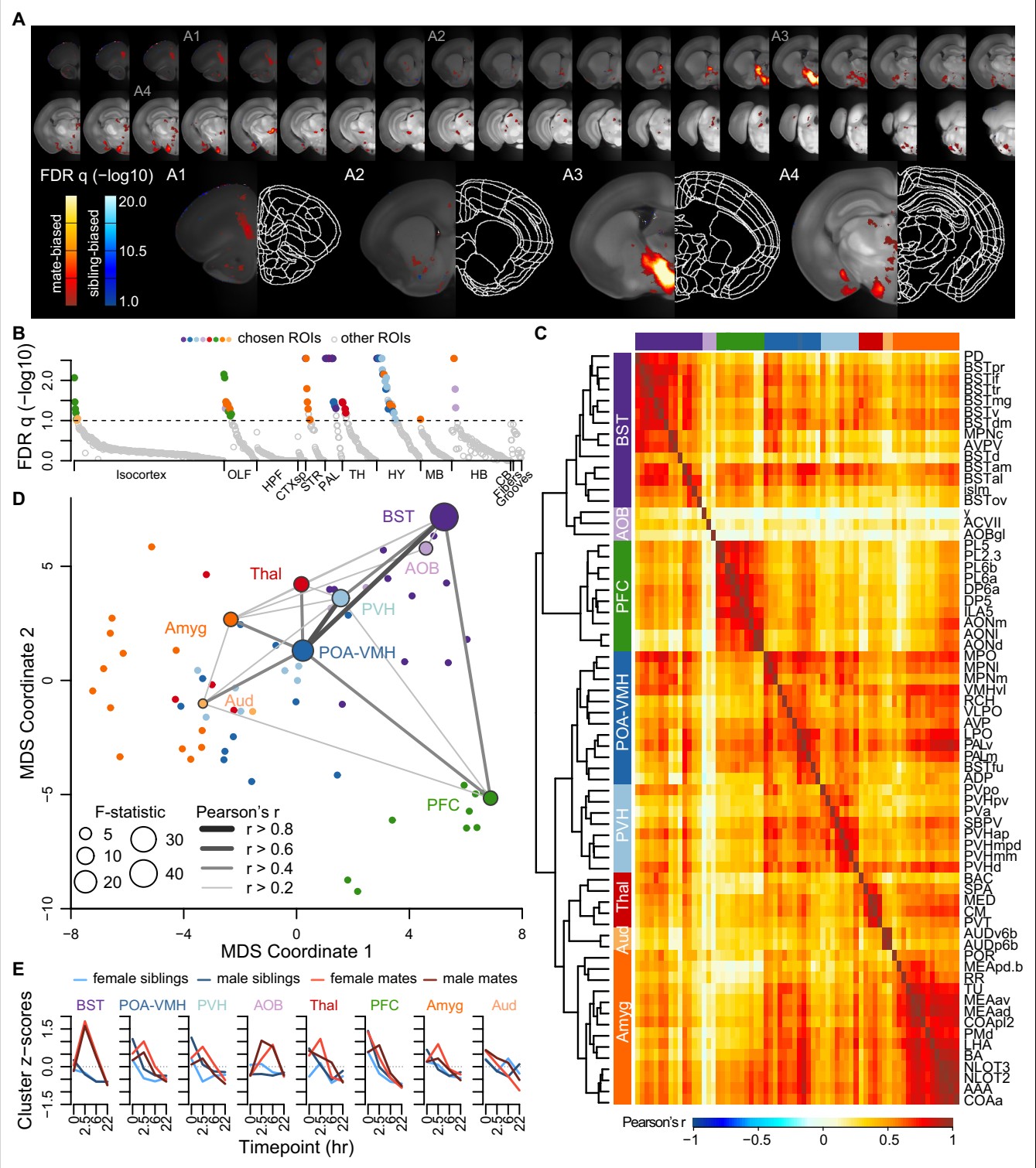

**Figure 3.** A brain-wide functional network is active during pair bond formation. (**A**) Map of bonding-associated voxels is shown in coronal sections (rostral to caudal) spanning the vole brain. Brightness indicates significance levels for comparisons of hypothesized and null generalized linear models (GLMs) to predict c-Fos+ cell counts across 189 vole brain samples. For each voxel, an analysis of variance (ANOVA) test was used to compare null and hypothesized GLMs. False discovery rate (FDR) $q$-values were computed from the ANOVA results to account for multiple comparisons across voxels (alpha threshold of q = 0.1). Warm colored voxels have higher mean c-Fos+ cell counts in mates (n = 94) compared to siblings (n = 95) (GLM test statistic for partner type > 0). Cool colored voxels have lower mean c-Fos+ cell counts in mates compared to siblings (GLM test statistic for partner type < 0). (**B**) Identification of significant and mutually exclusive regions of interest (ROIs, n = 68 chosen and 824 total ROIs) is shown sorted by anatomical division

*Figure 3 continued on next page*

*Figure 3 continued*

and *F*-statistic. For each ROI, an ANOVA test was used to compare null and hypothesized GLMs. The significance levels of ANOVAs were determined by comparing the observed *F*-statistic to its null distribution from shuffled data (10,000 permutations). False discovery rate (FDR) *q*-values were computed to adjust for multiple comparisons acros ROIs (alpha threshold of *q* = 0.1). Colored symbols for ROIs match their cluster group assignments in (**C**) and (**D**). (**C**) Hierarchical clustering of chosen ROIs (*n* = 68, permutation tests with FDR *q* < 0.1) is shown alongside pairwise Pearson correlations of c-Fos+ cell counts. Full ROI names are listed in **Supplementary file 4**. (**D**) Multi-dimensional scaling (MDS) coordinate space of the correlations is shown between chosen ROIs (*n* = 68). The most significant ROIs per cluster are labeled with the symbol size scaled by the *F*-statistic. Darkness and thickness of connecting lines reflect Pearson correlation coefficients. (**E**) Time course trajectories of total c-Fos+ cell counts are shown for each cluster. Counts per cluster are scaled across samples with means taken for each experiment group (*n* = 11-12 animals/group). Red lines represent mates and blue lines represent siblings (females – lighter, males – darker). In (**C**, **D**, **E**), each cluster group is given a label to summarize the most significant ROIs within it. Cluster group labels include 'BST' (bed nucleus of the stria terminalus) in dark purple, 'AOB' (accessory olfactory bulb) in light purple, 'PFC' (prefrontal cortex) in green, 'POA-VMH' (preoptic area, ventromedial hypothalamus) in dark blue, 'PVH' (paraventricular hypothalamus) in light blue, 'Thal' (thalamus) in red, 'Aud' (auditory cortex) in light orange, and 'Amyg' (amygdala) in dark orange.

The online version of this article includes the following figure supplement(s) for figure 3:

**Figure supplement 1.** Brain-wide patterns of immediate early gene activation during pairing.

**Figure supplement 2.** Patterns of immediate early gene expression in brain region clusters.

**Figure supplement 3.** Multi-dimensional structure of brain-wide correlation patterns.

**Figure supplement 4.** Anatomical connectivity in brain regions associated with pairing.

---

Following bonding (22 hr), within-pair correlations are confined to the coupling of activity between the hypothalamus and amygdala, including between medial preoptic area and medial amygdala (*Figure 4E*). In siblings, who were housed together since birth, hypothalamus–amygdala correlations are relatively uniform across timepoints, while correlations outside of these regions are largely absent (*Figure 4—figure supplement 3*).

The principal nucleus of the BST (BSTp), a subregion of the posterior BST, loads highly on CC1 (*Figure 4B*), and exhibits a strong response to mating in the 2.5 and 6 hr timepoints (permutation test: *n* = 189 animals, permutations = 10,000, FDR *q* = 0.003, *Figure 4C*); we found that this brain region also shows strong correlations in heterosexual pairs (Pearson correlation: df = 21, *r* = 0.851, p < 0.0001, *Figure 4G*). Remarkably, the number of observed male ejaculations is strongly predictive of BSTpr c-Fos+ cells in both males and females (Pearson correlation: males: df = 22, *r* = 0.894, p < 0.0001; females: df = 21, *r* = 0.847, p < 0.0001; *Figure 4H*). To our surprise, but consistent with the CCA, male ejaculation rates were the strongest predictor of activity across brain regions of both sexes (*Figure 4B, D*). Statistically controlling for the number of ejaculations effectively abolished the correlation we observed in mated pairs (*Figure 4F*).

## Discussion

Here, we present a brain-wide network of immediate early gene (IEG) circuits active as mating experience elicits a pair bond. We find no evidence of anatomical sexual dimorphism, and only modest evidence of dimorphic function. Our whole-brain mapping of pair-bond formation implicates 68 unique brain regions, including 18 regions that are within the primary brain network proposed for prairie vole bond formation (*Walum and Young, 2018*). The 68 identified regions are more strongly anatomically connected to one another than predicted by chance, and so can be interpreted as circuits active during pairing.

Although the majority of regions identified have not been linked directly to bonding, many are logical components of a pair-bonding network. For example, the strongest effects of pairing were detected within a cluster containing multiple compartments of the posterior part of the bed nucleus of the stria terminalus (BST), as well as the posterodorsal preoptic nucleus (PD), core medial preoptic nucleus (MPNc), and anteroventral periventricular nucleus ('BST' purple cluster, *Figure 3C–E* and *Figure 4A–C*). While this cluster exhibited the strongest response to pairing, only the BST has been previously implicated in prairie vole bonding (*Cushing and Wynne-Edwards, 2006*; *Lei et al., 2010*). The cluster as a whole, however, precisely matches the neural circuitry of male ejaculation previously mapped in rats, gerbils, and hamsters (*Coolen, 2005*; *Heeb and Yahr, 1996*; *Heeb and Yahr, 2001*; *Pfaus, 2009*; *Simmons and Yahr, 2002*). The posterior BST projects to both the medial preoptic nucleus (MPN) and the paraventricular nucleus (PVH), major contributors to two other clusters ('POA-VMH',

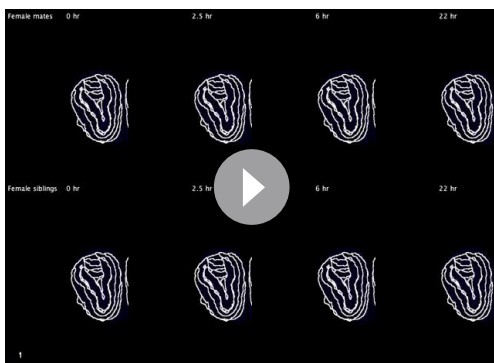

**Video 2.** Patterns of whole-brain immediate early gene expression during pairing. First, coronal cross-sections of immediate early gene (IEG) expression are shown for female (first video) and male (second video) prairie voles across all experiment groups split by partner type (mates – top, siblings – bottom) and timepoint (0, 2.5, 6, and 22 hr from left to right, *n* = 11-12 animals/group). Brightness corresponds to mean voxel c-Fos+ cell counts. These videos are associated with *Figure 3—figure supplement 1* (see this figure for the color key). The induction patterns are overlaid with prairie vole atlas boundaries. Next, coronal cross-sections are shown with the results of generalized linear model (GLM) comparisons to identify brain areas associated with bonding (*n* = 189 vole brain samples). This video is associated with *Figure 3A* (see this panel for the color key). The reference is overlaid with colored voxels that represent differences between hypothesized (includes partner type effect) and null (no partner type) GLMs. Analysis of variance (ANOVA) tests were used to compare GLMs, with a false discovery rate (FDR) correction across voxels (alpha threshold of *q* = 0.1). Warm colors indicate voxels with higher c-Fos+ cell counts in mate pairs (GLM test statistic for partner type > 0), and cooler colors indicate voxels with higher c-Fos+ cell counts in siblings (GLM test statistic for partner type < 0). Finally, coronal cross-sections are shown with the results of GLM comparisons to identify brain areas associated with sex differences. The reference is overlaid with colored voxels that represent differences between hypothesized (includes partner type x sex interaction) and null (no partner type x sex interaction) GLMs (ANOVA with FDR *q* < 0.1). Colored voxels indicate regions where there was an interaction effect, with warm colors indicating a positive interaction term and cool colors indicating a negative interaction term (see *Figure 3A* color key).

https://elifesciences.org/articles/87029/figures#video2

blue; 'PVH', light blue), regions that also exhibit strong responses to pairing (*Cushing et al., 2003*; *Insel and Young, 2001*). The paraventricular nucleus of the hypothalamus, moreover, is a major source of the neuropeptides oxytocin and vasopressin, known modulators of pair bonding in the prairie vole (*Walum and Young, 2018*).

A fourth cluster ('PFC', green) is composed of prelimbic, infralimbic, and olfactory cortex; activity in the vole prefrontal cortex is known to be modulated by hypothalamic oxytocin, and to shape bonding through projections to the nucleus accumbens (*Amadei et al., 2017*; *Burkett et al., 2016*; *Horie et al., 2020*). The pattern of activity in this cluster, however, indicates that it was due in part to differences between the isolated animals (0 hr) and other timepoints (*Figure 4—figure supplement 1* and *Figure 4—figure supplement 2*). Because animals in the isolated condition were in a compartment adjacent to either an opposite sexed individual or a familiar former cagemate, we cannot rule out that olfactory or auditory cues may have made animals aware of the presence of a potential social partner. Indeed, we interpret this dimension as capturing appetitive aspects of behaviors associated with investigation of the animal isolated from the subject by the barrier.

Although the prefrontal cortex (PFC) and other olfactory cortical areas formed a cluster, we did not find widespread c-Fos induction throughout the cortex in response to pairing. It seems likely that sensory and motor areas were important for social processes related to both pair-bonding and reunion with same-sex cagemates, such as investigation and recognition. Our study design, however, highlights differences between treatments, and in order to detect such effects, it might be necessary to compare mating and bonding pairs to animals left in complete isolation. Moreover, several cortical regions that did not survive corrections for multiple tests may have been identified in a less stringent analysis. Several subregions within the isocortex, hippocampal formation, and cortical subplate had statistical models that approached significance (i.e., p-values <0.1) prior to multiple test corrections. These subregions were found within primary somatosensory area, primary auditory area, dorsal and ventral auditory areas, primary visual area, anteromedial visual area, agranular insular area, temporal association areas, ectorhinal area, postsubiculum, and basomedial amygdala. Frontal cortex subregions were within the agranular insular area and orbital area, as well as additional subregions in prelimbic and infralimbic areas of the PFC.

In addition to its effects in the PFC, pairing drove increased c-Fos induction in the ventral pallidum, a major node in reward circuitry, as well as in the paraventricular nucleus and the medial preoptic area,

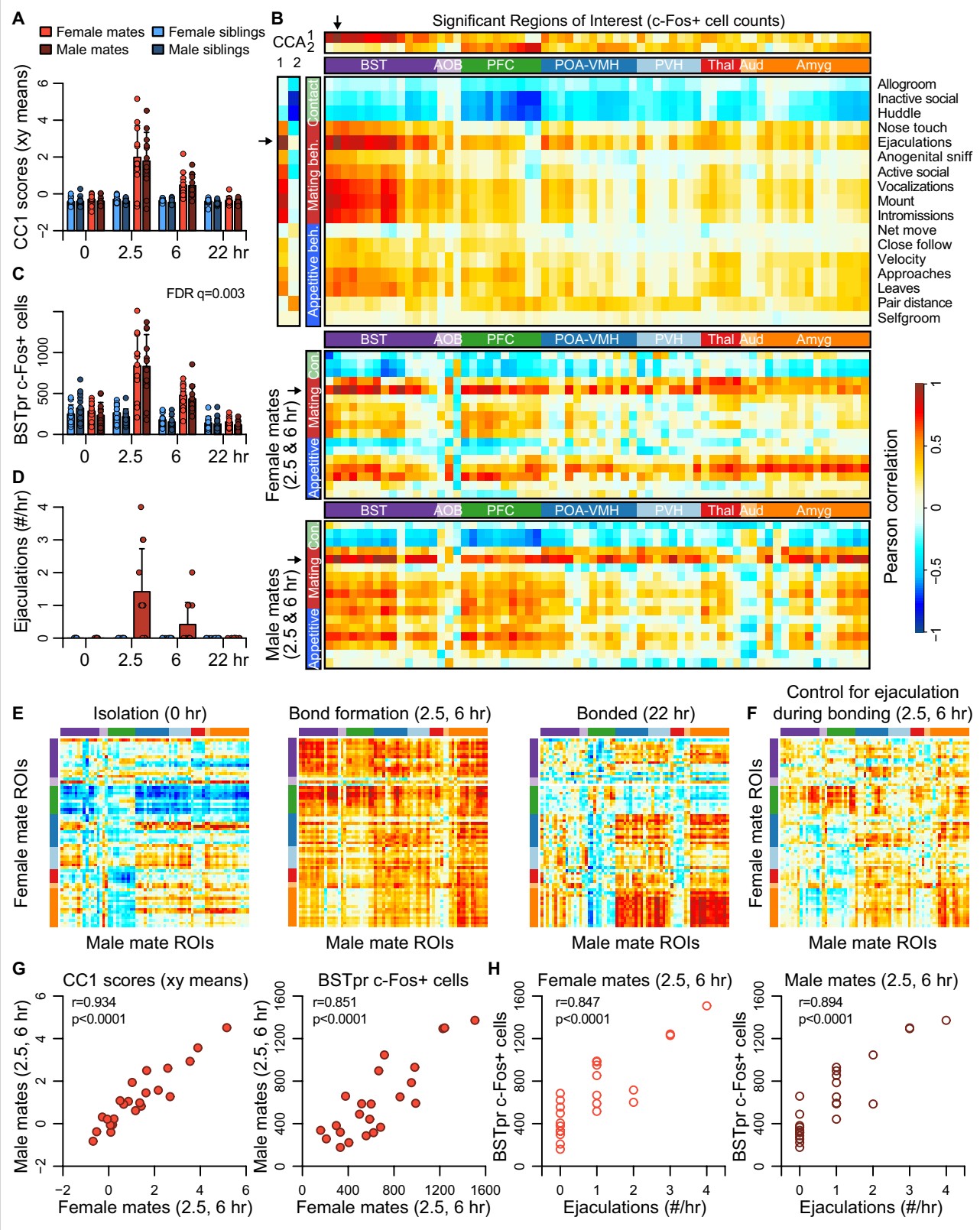

**Figure 4.** Bed nucleus of the stria terminalis (BST) emerges as a central hub in the bonding network that is associated with mating success in both sexes. (**A**) The first dimension of canonical correlation (CC) scores is compared across experiment groups (mean ± standard deviation [SD], $n$ = 11-12 animals/group). (**B**) Heatmaps represent correlation coefficients among CC scores, region of interest (ROI) c-Fos+ cell counts, and behavior measures. The full dataset is on top ($n$ = 189 animals for correlations involving ultrasonic vocalizations [USVs], $n$ = 187 for all other behaviors), and the bottom two

*Figure 4 continued on next page*

*Figure 4 continued*

correlograms are for female (*n* = 23) and male (*n* = 24) mates. In (**B**, **E**, **F**), warm and cool colors represent positive and negative coefficients, respectively. Arrows mark the ROI (principle nucleus of the BST [BSTpr]) and behavior (ejaculation) with the strongest correlations to CC1. (**C**) BSTpr activation is compared across groups (*n* = 11-12 animals/group) with a GLM comparison (ANOVA) followed by a permutation test (10,000 permutations) and false discovery rate (FDR) correction for multiple tests across ROIs. (**D**) Successful mating events are shown across timepoints (mean ± SD, *n* = 22-24 dyads/group). (**E**) Similarity (i.e., Pearson correlation) of bonding network activation is shown in female–male pairs at the 0 hr (*n* = 11), 2.5 and 6 hr (*n* = 23) and 22 hr (*n* = 12) timepoints. (**F**) Similarity in bonding network activation is shown in female–male pairs (*n* = 23), using partial Pearson correlations to control for ejaculation rates. (**G**) Female–male pair similarity is shown for CC1 scores (*n* = 24 pairs) and BSTpr activation (*n* = 23 pairs) during bond formation. (**H**) Pearson correlations are used to associate ejaculation rates with BSTpr activation for female mates (*n* = 23) and male mates (*n* = 24). In (**A**, **C**, **D**), mate pairs are in red and sibling pairs in blue (females in lighter hues, males in darker hues). In (**B**, **E**, **F**), cluster group labels include 'BST' (bed nucleus of the stria terminalus) in dark purple, 'AOB' (accessory olfactory bulb) in light purple, 'PFC' (prefrontal cortex) in green, 'POA-VMH' (preoptic area, ventromedial hypothalamus) in dark blue, 'PVH' (paraventricular hypothalamus) in light blue, 'Thal' (thalamus) in red, 'Aud' (auditory cortex) in light orange, and 'Amyg' (amygdala) in dark orange. ROI names and order correspond to *Figure 3C*. Full ROI names are listed in *Supplementary file 4*.

The online version of this article includes the following figure supplement(s) for figure 4:

**Figure supplement 1.** Dimensions of cross-covariance in immediate early gene expression and social behavior.

**Figure supplement 2.** Intra-pair similarity in immediate early gene expression in pairing-associated brain regions.

**Figure supplement 3.** Patterns of association between immediate early gene expression and behavior.

modulators of reward. This is consistent with a large body of work implicating neuropeptide actions on reward circuits in the formation of bonds (*Walum and Young, 2018*; *Young and Wang, 2004*). Conspicuously missing from our list, however, is significant pairing-induced c-Fos induction in the nucleus accumbens. The absence of significant accumbens IEG induction may reflect the limitations of using c-Fos and other IEGs as indicators of neural activity. It is known that some neuronal populations can be active without expressing c-Fos (*Sheng and Greenberg, 1990*). Indeed, although a variety of studies implicate the accumbens in bond formation (*Amadei et al., 2017*; *Aragona et al., 2006*; *Scribner et al., 2020*), previous work finds only weak c-Fos induction in the prairie vole accumbens during bonding (*Curtis and Wang, 2003*). Another possibility is that there was heterogeneous activation in the accumbens that was not captured by the precision of our atlas. Consistent with this interpretation, we found that the accumbens was significant in univariate tests, as well as in voxel-level analyses. Overall, our results do not conflict with pharmacological, electrophysiological, and calcium-imaging data on the role of the nucleus accumbens in prairie vole bonding (*Amadei et al., 2017*; *Aragona et al., 2006*; *Scribner et al., 2020*). Instead, the absence of significant effects at the level of the entire nucleus accumbens, together with the presence of anatomically restricted voxel-level significance, suggests substantial anatomical heterogeneity in the contributions of the nucleus accumbens to bond formation.

Our data suggest that pairing-related c-Fos immunoreactivity is shared across sexes, and that much of the pairing-related neural activity is driven by mating behavior. The strongest signal of pairing status was in the BSTpr, a region important in male ejaculation (*Insel and Young, 2001*; *Pfaus, 2009*). We found that both sexes exhibited the strongest signal of pairing status in BSTpr. This finding supports recent work in mice showing that different neural populations in BSTpr, aromatase+ and Vgat+ neurons, respond to ejaculation in both males and females (*Bayless et al., 2019*). Moreover, we found a surprisingly strong relationship between ejaculation rates and brain-wide activity patterns across the putative pair-bonding circuit. The concordance between sexes was surprising, but is consistent with IEG studies of ejaculation and related circuitry in male and female rats (*Coolen, 2005*; *Pfaus, 2009*). These data are also consistent with the interpretation that copulation enables coordination and assessment during bonding (*Carter et al., 1995*; *Dewsbury, 1988*; *Eberhard, 1996*; *Getz et al., 1993*). Our findings, along with this previous work, support the hypothesis that sexual behavior plays a key role in driving pair-bond strength. However, the current study focused on animals that were screened for sexual receptivity, which may have limited variation in sexual behavior across pairs. An intriguing direction for future research will be to test how this variation contributes to bond strength.

The tight coupling of widespread neural activity adds to the growing number of examples – including fish, mice, bats, and humans – that demonstrate correlated neural function across socially interacting individuals (*Hasson et al., 2012*; *Kingsbury et al., 2019*; *Kinreich et al., 2017*; *Long et al., 2020*; *Vu et al., 2020*; *Zhang and Yartsev, 2019*). Although widespread differences among mated pairs seem to be driven by ejaculation rates during mating and bonding, a smaller subset of

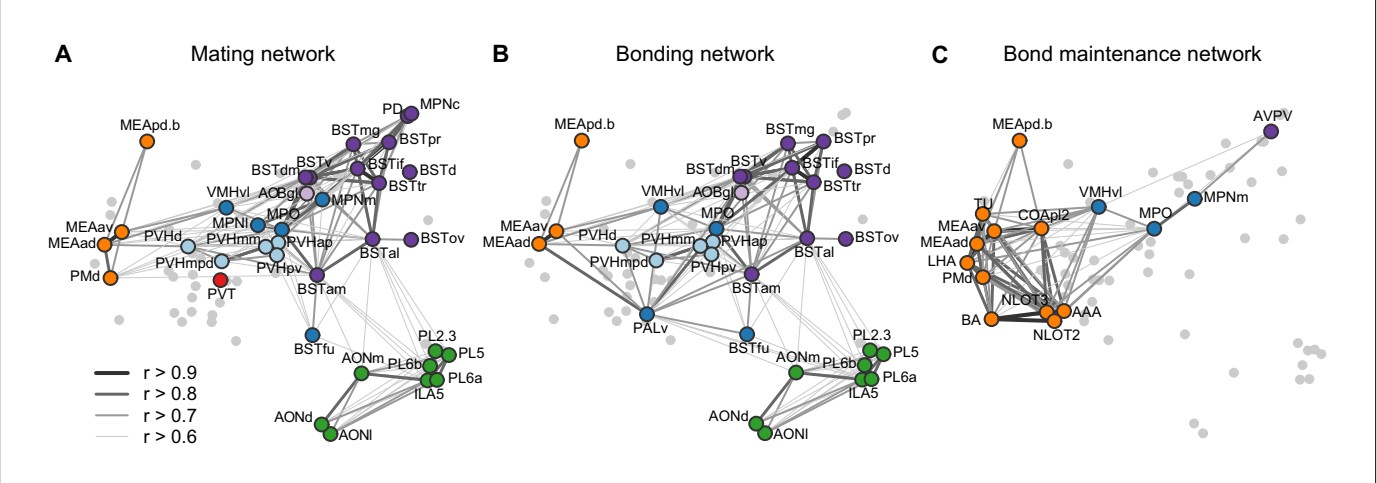

**Figure 5.** Working model for neural systems that shape stages of pair-bond development. (**A**) A schematic is shown of the network of regions identified in our study, overlaid with regions involved in rodent mating behavior (*Pfaus and Heeb, 1997*; *Veening et al., 2014*; *Veening and Coolen, 2014*). This network is proposed to be involved during the early stages of bond formation (e.g., 2.5 hr timepoint). (**B**) A schematic is shown of the network of regions identified in our study, overlaid with regions involved in prairie vole pair-bonding behavior (*Walum and Young, 2018*; *Young et al., 2011*). This network is proposed to be involved with the middle stages of bond formation when animals are engaged in prolonged mating and affiliative interactions (e.g., 6 hr timepoint). (**C**) A schematic is shown of the network of regions that are correlated between female and male mates at the 22 hr timepoint in this study. These regions are identified from pairs of regions in which both sexes show high inter-individual similarity (Pearson correlation, r > 0.75). This network is proposed to be involved with the recognition and convergence of behavioral state in bonding partners. The network schematics in (**A**–**C**) are adapted from the multi-dimensional scaling of correlations between region activity (*Figure 3D*). Light gray points represent regions not included in the proposed network. Line thickness and darkness between colored regions represent Pearson correlation r values between the connected regions. Cluster group labels include 'BST' (bed nucleus of the stria terminalus) in dark purple, 'AOB' (accessory olfactory bulb) in light purple, 'PFC' (prefrontal cortex) in green, 'POA-VMH' (preoptic area, ventromedial hypothalamus) in dark blue, 'PVH' (paraventricular hypothalamus) in light blue, 'Thal' (thalamus) in red, 'Aud' (auditory cortex) in light orange, and 'Amyg' (amygdala) in dark orange. Full ROI names are listed in *Supplementary file 4*.

circuits – specifically connections between the hypothalamus and amygdala – remain correlated after pairing. The similarity of this post-pairing pattern to non-sexual affiliative mechanisms recently documented in lab mice (*Hu et al., 2021*), and to correlations we observe among sibling pairs, suggests these regions play a role in both reproductive and non-reproductive attachment. These brain regions, and especially the amygdala, will be important candidates for future research on neural regulation of pair-bond maintenance.

Before offering a synthesis of our findings, it would be useful to acknowledge or reiterate a few caveats. First, as noted above, IEG induction does not capture all relevant neural activity (*Sheng and Greenberg, 1990*). Second, the design of our experiment, which controlled for social interaction, likely excluded many circuits important to both pair bonding and sibling social interactions. Third, c-Fos induction within a given brain region may nevertheless rely on distinct cell types, and so the absence of sex differences in c-Fos immunoreactivity does not definitively rule out the sexually dimorphic circuits hypothesized in the 'dual function hypothesis' (*De Vries, 2004*). Lastly, the current study focused on animals that were screened for sexual receptivity, which may have limited the variation in sexual behavior across opposite-sex pairs.

## Ideas and speculation

This brain-wide map of IEG induction provides a uniquely comprehensive perspective on the circuits that enable sexual behavior to become a bond. To help organize this enormous dataset, we looked to existing literature to compare the implicated circuits to those related to sociosexual behaviors in other species (*Figure 5A, B*). Overlaying the time course of behaviors and circuit activity suggest distinct stages of bonding and corresponding neural function. We divide those stages into mating, bonding, and ongoing bond maintenance.

The first canonical correlate captured the major dimensions of both brain and behavioral variation, and showed the largest group differences surrounding the 2.5 hr timepoint. Behaviorally, this difference mapped strongly onto mating behavior, and in particular to rates of male ejaculation. The neural

circuits of rodent sexual behavior in general, and of male ejaculation in particular, have been well studied in rats, hamsters, and laboratory mice (*Bayless et al., 2019*; *Pfaus and Heeb, 1997*; *Veening et al., 2014*; *Veening and Coolen, 2014*). In this context, chemosensory signals are thought to drive activity in medial amygdala (MeA), medial preoptic area (MPO), and PVH. Copulation drives activity in ventromedial hypthalamus (VMH), MPN, PD, and BST. We found that elevated c-Fos induction levels occur in these areas early on in the bonding time course and then decline. Such findings support the idea that these core circuits orchestrate early social interactions that enable individuals to process chemosensory signals and then initiate lordosis, mounting, and ejaculation. Our study also revealed a set of regions that have yet to be emphasized in sociosexual circuits, including lateral hypothalamic area and lateral preoptic area. While research on these regions emphasize non-social functions, a number of studies suggest that these regions play a role in social behaviors, or participate in goal-directed behavior and reward (*Chen et al., 2021*; *Nieh et al., 2016*; *Petzold et al., 2023*).

Interestingly, one anomaly from the literature on sexual behavior circuits is that females also show IEG induction associated with male ejaculation – a finding evident in both rats and mice (*Pfaus and Heeb, 1997*; *Veening et al., 2014*). Similarly, recent work in female laboratory mice demonstrate that Vgat+ neurons in the BSTpr exhibit responses to male ejaculation (*Bayless et al., 2019*). The extraordinary pattern of coordination we observe between members of a pair across nearly 70 brain regions, and its prediction by male ejaculation, suggest that both females and males are experiencing similar and profound affective states. Moreover, the fact that repeated copulation is necessary for bonding is consistent with work in a variety of taxa suggesting that repeated copulation is a means of partner assessment (*Dewsbury, 1988*; *Eberhard, 1996*); in prairie voles, we suggest that both females and males are assessing the ability of a male to monopolize a female, a trait that would predict male paternity and ability to defend a nest against conspecifics in the field. Pfaus et al. have argued that females of other species, including laboratory rodents, exhibit orgasm-like responses (*Georgiadis et al., 2012*). Although our current data are unable to address this claim directly, the hypothesis offers a parsimonious interpretation of our data, and the topic merits further scrutiny.

In this circuitry, the 'extended amygdala' regions of the BST and MeA stand out for their extensive projections to and from the hypothalamus, and for their known roles in individual recognition. *Dumais et al., 2016*, for example, have documented that the BST of the rat is essential to individual recognition; in parallel, *Knoedler et al., 2022* find that estrogen receptor alpha (ERa) in the BST coordinates sex recognition in laboratory mice. Most directly, Cushing et al. have shown that ERa in the BST is essential to the formation of pair bonds in prairie voles (*Lei et al., 2010*). Similarly, oxytocin actions in the MeA are known to govern social recognition in lab mice (*Ferguson et al., 2001*). We hypothesize that IEG induction in response to mating allows the identity of a mate to get access to the modulation of hormonal and behavioral states by the hypothalamus. Such identity representations may coordinate partner-specific patterns of neuropeptide release (PVN), selective aggression (ventrolateral VMH [VMHvl]), selective mating (VMHvl, MPO), and social reward (MPO).

After bond formation and stabilization, we find that that mated pairs are similar to co-housed siblings in terms of brain-wide IEG induction patterns (*Figure 5C*). More remarkable, is that after 22 hr both co-housed siblings and mated pairs show strong correlations in amygdala IEG induction, particularly in the MeApd. We propose that this correlated activity reflects a sensitivity to the behavioral state of a partner that emerges as a function of bonding. Recent work has revealed that in laboratory mice, tachykinin + GABAergic neurons project from the posterodorsal MeA (MeApd) to the MPO, where they contribute to both consolation grooming and social reward (*Hu et al., 2021*; *Wu et al., 2021*). These projections have not been implicated in pair bonding, but our current data suggest that this circuit may play a broader role in the ongoing function of relationships in both mates and siblings. Since prairie voles often live in larger family groups (*Getz et al., 1993*), particularly over winter, a common mechanism for the maintenance of familial and sexual bonds would be consistent with their ecology. In addition, the strong correlation of activity across a variety of related amygdala and associated regions suggest this circuit may be anatomically much broader than the interactions between MeApd and MPO documented in laboratory mice.

## Conclusions

Overall, our data survey the brain to identify circuits active as mating produces a pair bond. These data allowed strong tests of opposing hypotheses about sexual dimorphism and coordination during

bonding (*De Vries, 2004*; *Hasson et al., 2012*; *Kingsbury et al., 2019*; *Kinreich et al., 2017*; *Long et al., 2020*; *Zhang and Yartsev, 2019*), revealing a surprising absence of sexual dimorphism in structure and function, and an extensive coordination of neural activity among new pairs. We confirm the activity of known regulators of mating and bonding such as reward circuits, the paraventricular nucleus, and the bed nucleus of the stria terminalis (*Coolen, 2005*; *Insel and Young, 2001*; *Lei et al., 2010*; *Pfaus, 2009*); we map a path by which sexual activity finds its way to known regulators of bonding, and identify a variety of novel regions and circuits whose roles in bonding remain to be examined. Our results suggest a novel model in which the BST is a key node connecting sexual experience to the neuroendocrine functions of the hypothalamus and preoptic area, and that coupling between the preoptic area and the amygdala may play an unappreciated role in the maintenance of established bonds. Manipulations of these circuits and their behavioral consequences offer rich new opportunities for research into the mechanisms of bonding and their contributions to well-being.

## Materials and methods

### Animals

Prairie voles (*Microtus ochrogaster*) used in behavioral and anatomical experiments were derived from wild-caught voles from Jackson County, Illinois and bred at The University of Texas at Austin. At weaning (PND 21), voles were housed in polycarbonate cages (R20 Rat Cage, Ancare Corp, NY) in groups of two to five same-sex littermates and provided with standard rodent chow (LabDiet 5001, Lab Supply, TX) and water ad libitum. The colony room was maintained at a controlled temperature (20–23°C), and the photoperiod was on a 12:12 light:dark cycle (lights on: 0600, lights off: 1800). Housing arrangements allowed animals to receive visual and olfactory cues, but not tactile contact, with conspecifics. Mice used for anatomical research were housed with ad libitum access to food and water in a controlled temperature (21–22°C) and light (12:12 light/dark cycle) room. All animal procedures were approved by the Institutional Animal Care and Use Committees (IACUC) at the University of Texas at Austin (protocols #2016-00247 and #2019-00228) and Cold Spring Harbor Laboratory (protocol #18-15-12-09-12).

### Brain tissue sample preparation and processing

Prairie voles underwent intracardiac perfusion ~20 min (mean = 21 min, range = 10–40 min) after the behavioral observation endpoints (2 hr acclimation or 2 hr acclimation in addition to 2.5, 6, or 22 hr cohabitation). Animals were anesthetized with open-drop isoflurane exposure, exsanguinated with 0.9% saline (2 min 30 s at 13 ml/min), and fixed with 4% paraformaldehyde (PFA) in 0.05 M phosphate buffer (PB; 5 min 30 s at 13 ml/min). All harvested brain tissue samples were post-fixed overnight at 4°C in 4% PFA in PB. After post-fixation, samples were washed 3× in 0.05 M PB and stored in 0.05 M PB 0.02% sodium azide at 4°C until immunolabeling processing.

Mice of 8–10 weeks old were anesthetized with ketamine/xylazine and transcardially perfused with isotonic saline followed by 4% PFA in 0.1 M PB (pH 7.4). Brains were extracted and post-fixed overnight at 4°C in the same fixative solution, and stored at 0.05 M PB until immunolabeling processing.

### Whole-brain IEG staining and imaging

The right hemisphere of all brain samples were cut and immunolabeled for immediate early gene (IEG) c-Fos and posteriorly cleared using iDisco+ protocol (*Renier et al., 2014*; *Renier et al., 2016*). Briefly, samples were initially delipidated with methanol and later permeabilized and blocked with DMSO (dimethyl sulfoxide) and donkey serum, respectively. Thereafter, all samples were incubated with c-Fos antibody and Alexa Fluorophore 647 secondary antibodies. Samples were cleared with increasing concentration steps of methanol and dichloromethane as previously described (*Renier et al., 2014*; *Renier et al., 2016*). After clearing, samples were imaged sagittally on a light-sheet fluorescence microscope (Ultramicroscope II, LaVision Biotec). Samples were imaged continuously every 5 μm at 640 and 488 nm, for signal and background channels, respectively.

### Construction of the vole reference brain and atlas

For the construction of the prairie vole reference brain, 190 brains from c-Fos+ cell counting analysis were co-registered as described below (3D registration of the vole brain) (*Kim et al., 2015*). For the

construction of the prairie vole reference atlas, the Allen Reference Atlas (ARA) was initially used as a template. After initial registration, output transformations were used to warp mouse atlas onto the prairie vole reference brain. In order to validate atlas registration, prairie vole Neurotrace stained brains were registered at high resolution onto the prairie vole reference brain.

## 3D registration of the vole brain

Brains were registered to a standardized reference brain as previously described (*Renier et al., 2014*; *Renier et al., 2016*). Initial 3D affine transformation was calculated using six resolution levels followed by a 3D B-spline transformation with three resolution levels. Similarity was computed using Advanced Mattes mutual Information metric by Elastix registration toolbox. In order to enhance image registration, both brain images and reference brains were pre-processed to reduce the impact of imaging artifacts during the computation of mutual information (*Video 1*). First, brain image illumination was corrected to homogenize illumination across sections. Second, both brain images and reference brain intensities were smoothed to reduce imaging artifacts (Muñoz-Castañeda and Osten, In Preparation).

## Whole-brain prairie vole Neurotrace staining

Whole-brain Neurotrace staining was performed with a modification of the iDISCO+ protocol (*Muñoz-Castañeda et al., 2021*; Muñoz-Castañeda and Osten, In Preparation). Samples were initially washed in phosphate-buffered saline (PBS) and incubated for in PBS + Triton X-100 + DMSO + glycine. Then samples were transferred and incubated in a solution with PBS and Neurotrace. Finally, samples were washed in PBS.

## Comparison of vole and mouse neuroanatomy

For area-based volume quantifications, the reference brain was registered onto each single imaged brain and all volume areas were automatically quantified with custom made scripts (Muñoz-Castañeda and Osten, In Preparation). These area volumes were used for comparisons within and between species ($n$ = 96 female voles, 94 male voles, and 108 male mice).

## STPT whole-brain imaging for anatomical delineation

Before imaging, both prairie vole and mouse brains were embedded and cross-linked with oxidized 4% agarose as previously described (*Kim et al., 2017*; *Ragan et al., 2012*). Whole-brain imaging was achieved using the automated whole-mount microscopy with serial two-photon tomography (STPT). The entire brain was coronally imaged at an *X,Y* resolution of 1 µm and *Z*-spacing of 50 µm (*Kim et al., 2017*; *Ragan et al., 2012*). After imaging, brains were registered to the reference brain for anatomical validation (see 3D registration of the vole brain).

## Automated c-Fos+ cell detection

c-Fos+ cells segmentation was performed using convolutional neural networks as previously described (*Kim et al., 2016*; *Kim et al., 2017*). After cell segmentation, all cell centroids were calculated for whole-brain distribution analysis. For the analysis of area-based c-Fos+ cell counts, the mouse reference brain was registered onto each prairie vole brain, using the output transformations to warp the atlas onto each individual brain for region of interest (ROI) cell counting and distribution. For voxel analysis, the inverse transformation output was used to move all c-Fos centroids onto the reference brain.

## Experiment design

Study subjects were 8–12 week old sexually naive prairie voles. There were eight treatments that varied based on partner type and cohabitation time (*Figure 2*). Subjects were partnered with a familiar same-sex cage mate (siblings) or an opposite-sex individual (mates) for 0, 2.5, 6, or 22 hr. The 0 hr timepoint represents a baseline state before pairing takes place, the 2.5 hr timepoint is ~2 hr after the first mating bout, the 6 hr timepoint is when an unstable partner preference is established, and the 22 hr timepoint is when pair bonding becomes stable (i.e., after overnight mating) (*DeVries and Carter, 1999*; *Williams et al., 1992*). It is important to note that the opaque divider in the acclimation period prevented physical interactions, but it is possible that animal pairs may have detected each other through olfactory or auditory cues. The experiment was run in six testing blocks. Each

block was composed of eight testing days spread across 2 weeks, with two adjacent testing days per cohabitation timepoint (0, 2.5, 6, and 22 hr). Two pair sets were tested in a day, either female siblings and male siblings or two mating pairs. The order of partner type (adjacent days per timepoint) and order of cohabitation timepoints (four per block) were counterbalanced across blocks. Individuals were randomly assigned to the partner type condition.

There were 12 sibling pairs and 12 mating pairs tested for each cohabitation timepoint, for a total of 96 female and 96 male study subjects. Of this total, 190 voles were used for behavioral analyses, and 189 voles were used for IEG analyses. A mating pair from the 0 hr timepoint group lacked behavioral data due to a camera malfunction (audio was unaffected). Three brain samples were not used in IEG analyses: one male sibling from the 6 hr timepoint group, one female mate from the 0 hr timepoint, and one female mate from the 6 hr timepoint. The first sample was not stained for c-Fos due to major issues with the perfusion. The latter two samples were identified as outliers by a Rosner's test (EnvStats R package; *Millard, 2013*); their whole-brain c-Fos+ cell counts were higher than the rest of the samples ($R$ = 4.61 and 5.504, p < 0.05).

## Estrus induction and animal selection

Voles were housed in new home cages, isolated from their sibling cage mates, 4–5 days prior to testing. During this isolation period, all females were induced into estrus with daily 0.1 ml subcutaneous injections of 2 µg estradiol benzoate dissolved in sesame oil (*Amadei et al., 2017*; *Carter et al., 1988*). Voles were screened for mating capacity on the third day of isolation. This screening involved a brief exposure (<10 min) to an opposite-sex vole until the first mount. Voles that did not show a mating attempt in the first exposure were retested with a different animal. We used this mating assay to restrict study subjects to voles that showed lordosis (females) or mounting behavior (males). By selecting voles who showed sexual behavior, we could control the estrus state and timing of mating across the 0, 2.5, 6, and 22 hr study groups. This selection process also ensured that animals assigned to the same-sex sibling pair and opposite-sex mating pair groups had similar sexual motivation and experience.

## Behavioral procedures

Behavioral testing began in the morning after lights-on between 8:00 and 10:15 (mean = 8:44 am). Subjects were fitted with colored collars (pipe cleaner attached to a miniature cable tie) for identification and automated video tracking. Then, subjects were placed on opposite sides of a custom made acrylic testing arena (12″ × 24″ × 12″), separated by an opaque divider, for a 2 hr acclimation period. The testing arenas were outfitted with fresh bedding and ad libitum access to food and water. The arenas were housed within an enclosed experiment box (42″ × 34″ × 24″) constructed from expanded PVC. These experiment boxes had controlled white:red lighting (Phillips Hue light strips) on the same photoperiod cycle as the colony room. For the 0 hr timepoint, subjects only underwent a 2 hr acclimation period. Otherwise, the divider was removed at the end of acclimation so that subjects could interact freely for 2.5, 6, or 22 hr. After this acclimation (0 hr group) or cohabitation (2.5, 6, and 22 hr groups), subjects were promptly removed from the arena and perfused. Arenas were cleaned with 70% ethanol between tests.

Video and audio data were recorded from Basler Ace-IMX174 cameras (1920 × 1200, 2 MP resolution, Basler AG, PA) and Ultramic 364K BLE microphones (Dodotronic, Italy) suspended above each testing arena. Cameras were outfitted with a 16-mm lens and 46-mm linear polarizer. Video data were recorded with Pylon viewer (v 5.1.0.12681, Basler AG, PA) at 25 fps with white balance set to ~1.93 and exposure levels between 15 and 25 ms. Audio was recorded with SeaPro2 software (v 2.0j, CIBRA) at a 192-kHz sampling rate and saved in 30 min WAV files (starting on the hour and every half hour). With this setup, ultrasonic vocalizations (USVs) could be detected per pair, but not localized to specific individuals. Recording equipment was connected directly to a Windows 10 PC. The start of both video and audio recordings were timestamped with PC system time, which enabled the synchronization of audio and video data on a second-by-second timescale.

## Video and audio processing

Behaviors were measured during observation windows that corresponded to peaks in c-Fos induction (*Kim et al., 2015*; *Renier et al., 2016*), specifically 60–120 min before each pair's perfusion (i.e.,

midpoint between exsanguination times of each vole pair). Both automated and manual methods were used to characterize vole behavior.

Automated scoring of movement and proximity behaviors was done with Ethovision (v 10.1). Individual collars were tracked during white light periods, with detection settings set for each video and optimized for collar color (blue/green, mean ± standard deviation [SD]: huemin = 97.0 ± 1.4/63.1 ± 2.5, huemax = 113.0 ± 1.4/79.2 ± 2.6, saturationmin = 113.3 ± 11.8/85.4 ± 10.5, saturationmax = 255.0 ± 0.2/254.9 ± 0.7, brightnessmin = 84.1 ± 10.5/70.2 ± 8.9, brightnessmax = 254.0 ± 4.6/254.7 ± 1.6, marker size = 50). Body area was recorded across both white light and red light periods with the grayscaling method (white/red light: detectionmin = 0, detectionmax = 108.7 ± 3.3/5.96 ± 0.8, pixel sizemin = 3000, pixel sizemax = 125,000, contour erosion = 1, contour dilation = 1). The largest body area was recorded when animals were in physical contact. Overall activity (% pixel change between frames) also was recorded for white light and red light periods separately (white/red light: threshold = 1, background = 10, compression = 'on' for all 24 overnight videos/'on' for 18 overnight videos). Individual tracking, body area, and activity were recorded at 12.5 samples/s, down sampled to 1/s, and used to compute an array of behavioral measures across entire recording sessions and the 1 hr observation windows (*Figure 2—figure supplement 1* and *Supplementary file 3*).

Manual scoring of mating, investigative, grooming, and contact behaviors was done with BORIS (v 7.9.6) (*Friard et al., 2016*). Two trained observers, who were blind to animal sex and partner condition, independently labeled social and non-social behaviors in cohabitation videos (2.5, 6, and 22 hr groups) during the 1 hr observation windows (*Supplementary file 3*). Self-grooming during acclimation videos (0 hr group) was labeled by one of the trained observers. Inter-observer reliability was assessed with Pearson correlations. For dyadic behaviors (e.g., mounting and huddling), reliability was assessed across vole pairs. For individual behaviors (e.g., anogenital sniffing and self-grooming), reliability was assessed separately for individuals with blue collars and green collars. All social behaviors had high agreement between the independent observers (Pearson correlations, $r > 0.90$; *Supplementary file 3*).

Ultrasonic vocalizations (USVs) were detected with DeepSqueak (v 2.6.1 in MATLAB 2019a) (*Coffey et al., 2019*). To initially label USVs, the 'AllShortCalls' network was used along with the following detection settings: frequency range of 20–90 kHz, overlap of 0.1 s, 5 s chunk length, and a high precision recall. Then, a trained observer revisited all USVs labels during the 1 hr behavior observation windows. Using the DeepSqueak interface, the observer removed labels of background noise (e.g., water bottle sounds) and adjusted USV label boundaries to exclude noise. This trained observer was blind to the partner condition, caller sex, and behavioral context. The start and stop times of all USV labels were exported and the times were adjusted to align with the video frame timestamps.

## Statistical analyses

Absolute and relative volumes of brain regions were compared across species (mouse vs. vole) and sexes (male voles vs. female voles) with negative binomial regression models. Relative volumes were computed as the ratio of region volumes to whole-brain volumes. To control for multiple comparisons, p-values were corrected with the False Discovery Rate (FDR) method (*Benjamini and Hochberg, 1995*).

All statistical analyses for behavioral experiments were carried out in R (v 3.5.3) (*R Development Core Team, 2016*). Welch two-sample *t*-tests were used to compare behavioral measures between mate pairs and sibling controls. Paired *t*-tests were used to compare behavioral measures between female mates and male mates. *T*-tests were run for all timepoints combined and for specific timepoints. Significance values were adjusted for multiple comparisons using the FDR method, with an alpha threshold of $q = 0.05$. Pearson correlations were used to make associations between behavioral measures and ROI c-Fos+ cell counts.

To identify brain regions that are sensitive to pairing status, we used a model comparison approach. For each voxel or ROI, two general linear regression models ('GLMs'; a quasi-Poisson link function was used to model over-dispersed count data) were used to predict IEG c-Fos expression across all individuals ($n = 189$) (*Figure 3*, *Video 2*). These models differed in whether the partner condition (sibling vs. mate) was a part of the formula. The 'null' model (*Equation 1*) included main effects of sex [S], timepoint [T], and experiment block [B]. The hypothesized 'bonding' model (*Equation 2*) included the same predictor terms in addition to a main effect of partner type [P] and interactions between partner

with sex or timepoint. In these models, sex was a categorical factor (males vs. females). Timepoint was an ordinal variable from 1 to 4 (0, 2.5, 6, and 22 hr) and was included as a polynomial term to account for both linear and quadratic (i.e., 0/22 vs. 2.5/6 hr) effects. Block was an ordinal variable from 1 to 6. Partner was a categorical variable (mates vs. siblings). This model comparison allowed for the identification of voxels/ROIs where c-Fos induction variation was explained better when accounting for the partner condition.

$$Y_i = \beta_1 S_i + \beta_2 T_i + \beta_3 T_i^2 + \beta_4 B_i + e_i \tag{1}$$

$$Y_i = \beta_1 S_i + \beta_2 Ti + \beta_3 T_i^2 + \beta_4 B_i + \beta_5 P_i + \beta_6 P_i^* S_i + \beta_7 P_i^* T_i + \beta_8 P_i^* Y_i^2 + e_i \tag{2}$$

The null and bonding GLMs were compared with an analysis of variance (ANOVA) test. For the ROI-level analysis, a Monte Carlo permutation approach (10,000 random shuffles of the data) was used to determine a null distribution for the ANOVA *F*-statistics and compute p-values. For the voxel- and ROI-level analyses, the FDR method was used to correct for multiple tests, with an alpha threshold of *q* = 0.1.

We ran an additional model comparison to specifically test for sex differences in c-Fos induction (*Video 2*). In this analysis, we compared the full version of the model formula (*Equation 2*) to a reduced version of the model (*Equation 3*). The reduced model excludes a sex by partner interaction term. The FDR method was used again to correct for multiple tests across all voxels and all ROIs, with an alpha threshold of *q* = 0.1.

$$Y_i = \beta_1 S_i + \beta_2 T_i + \beta_3 T_i^2 + \beta_4 B_i + \beta_5 P_i + \beta_6 P_i^* T_i + \beta_7 P_i T_i^2 + e_i \tag{3}$$

ROIs were chosen for subsequent analyses if they surpased the FDR alpha threshold (*q* = 0.1) in *Equation 1* vs. *Equation 2* model comparison and were mutually exclusive with one another. To select mutually exclusive ROIs, we constructed an structural hierarchy from the ARA. For any significant ROIs that overlapped anatomically, we iteratively chose the ROI with the higher ANOVA *F*-statistic. This method excluded larger brain regions in lieu of smaller and more localized ROIs.

Significant 'chosen' ROIs were assigned to groups by using hierarchical clustering with the ward D2 method, with a Euclidean distance matrix extracted from c-Fos+ cell counts (*Murtagh and Legendre, 2014*; *R Development Core Team, 2016*). The resulting tree was cut so that ROIs were grouped into anatomically similar clusters. We used multi-dimensional scaling (MDS) to further interpret the degree of similarity between chosen ROIs and their clustering groups based on the Euclidean distance matrix. The MDS method is a form of non-linear dimensionality reduction to visualize similarity in Cartesian space (*Cox and Cox, 2008*).

To confirm whether chosen ROI clusters represented structural circuits, we compared cluster assignments to published data on structural connectivity in the mouse brain (*Knox et al., 2019*). First, we refined a matrix of ROI–ROI ipsilateral normalized connection densities to align with our list of chosen ROIs. Some of our chosen ROIs did not align with this matrix because they represented subregions of the ROIs in the matrix. In those cases, we used data from the next inclusive ROI that was available (e.g., BST data used for BSTpr). Then, for each cluster, we found the mean normalized connection density between the regions. We excluded data from the matrix diagonals (e.g., BST to BST) to emphasize connections between, rather than within, regions. We took an average of these cluster densities values to capture the overall connection density based on our cluster assignments. A permutation test was used to assess whether this connection density was higher than expected by chance. The rows of the connectivity matrix (rows = origin ROIs, columns = target ROIs) were randomly shuffled prior to computing the mean cluster density, and this was done 10,000 times to construct a null distribution. This null distribution was then compared to the observed density to estimate its probability.

We used canonical correlation analysis (CCA) to investigate the relationships between behavioral variables and IEG induction patterns in chosen ROIs. CCA is an unsupervised method that finds linear combinations of two variable sets with the strongest correlation (*González et al., 2008*; *R Development Core Team, 2016*; *Wang et al., 2020*). This approach enabled us to isolate discrete canonical

correlates (CC), where each individual animal is assigned scores for each variable set per CC factor. We used correlations between individual behavior/ROI measures and their CC scores to interpret CC factors and identify specific behaviors and ROIs with the strongest associations. Wilk's lambda test statistic was used to confirm which CC factors represented a significant association between the two variable sets.

## Acknowledgements

We thank animal resource staff at the University of Texas at Austin and Cold Spring Harbor Laboratory for their assistance with animal care and husbandry. Tracy Burkhard helped with animal perfusions and tissue collection, and Desiree Lama and Morgan Klein assisted with behavior scoring. We thank Janelle Collins and other staff at Certerra Inc for their assistance with immunohistochemistry, whole-brain imaging and preliminary analyses, and we thank Kith Pradhan for advice on customizing the voxel and ROI analysis pipeline. Rhonda Drewes and Jason Palmer assisted with alignment of the reference atlas to the prairie vole brain and with imaging. Finally, we thank Drs. Hans Hofmann. Adam Kepecs, Micheal Long, and Michael Ryan for their feedback on earlier versions of this manuscript. This research was funded by grants from the National Institutes of Health: R01MH115267 (S.M.P., P.O.) and K99MH126164 (M.L.G.).

## Additional information

### Competing interests

Pavel Osten: P.O. is a co-founder, shareholder and Director at Certerra, Inc and co-founder, shareholder and President at Certego Therapeutics, Inc. The other authors declare that no competing interests exist.

### Funding

| Funder | Grant reference number | Author |
| --- | --- | --- |
| National Institute of Mental Health | R01MH115267 | Steven M Phelps<br>Pavel Osten |
| National Institute of Mental Health | K99MH126164 | Morgan L Gustison<br>Steven M Phelps |

The funders had no role in study design, data collection, and interpretation, or the decision to submit the work for publication.

### Author contributions

Morgan L Gustison, Conceptualization, Data curation, Formal analysis, Investigation, Visualization, Methodology, Writing – original draft, Project administration, Writing – review and editing; Rodrigo Muñoz-Castañeda, Conceptualization, Data curation, Formal analysis, Investigation, Visualization, Methodology, Writing – review and editing; Pavel Osten, Conceptualization, Supervision, Funding acquisition, Methodology, Writing – review and editing; Steven M Phelps, Conceptualization, Supervision, Funding acquisition, Methodology, Writing – original draft, Writing – review and editing

### Author ORCIDs

Morgan L Gustison ⓘ https://orcid.org/0000-0002-1162-8966
Rodrigo Muñoz-Castañeda ⓘ http://orcid.org/0000-0002-1176-7421
Pavel Osten ⓘ https://orcid.org/0000-0002-6385-7541
Steven M Phelps ⓘ http://orcid.org/0000-0002-1095-361X

### Ethics

All animal procedures were approved by the Institutional Animal Care and Use Committees (IACUC) at the University of Texas at Austin (protocols #2016-00247 and #2019-00228) and Cold Spring Harbor Laboratory (protocol #18-15-12-09-12).

Reviewer #1 (Public Review): https://doi.org/10.7554/eLife.87029.3.sa1
Reviewer #2 (Public Review): https://doi.org/10.7554/eLife.87029.3.sa2
Reviewer #3 (Public Review): https://doi.org/10.7554/eLife.87029.3.sa3
Author Response https://doi.org/10.7554/eLife.87029.3.sa4

## Additional files

### Supplementary files

• Supplementary file 1. Statistical comparisons of absolute and relative brain area volumes between prairie voles and mice. Absolute and relative area volumes (normalized to total brain volume) are compared between prairie voles ($n$ = 190 voles, 94 males and 96 females) and male mice ($n$ = 108) with negative binomial regression models. Absolute brain area volumes are all statistically significant bigger in the prairie vole compared to the mouse, but none the ratios of area volume (relative to the whole brain) were statistically different between species. Mean, standard deviation (SD), p-values, and false discovery rate (FDR) correction of the p-values are presented for each region.

• Supplementary file 2. Statistical comparisons of absolute and relative brain area volumes between male and female prairie voles. Absolute and relative area volumes (normalized to total brain volume) are compared between 94 male and 96 female prairie voles with negative binomial regression models. No statistical differences were found in these sex comparisons. Mean, standard deviation (SD), p-values, and false discovery rate (FDR) correction of the p-values are presented for each region.

• Supplementary file 3. Prairie vole behavior ethogram. Descriptions are included for automated, semi-automated, and manually scored behavioral measures.

• Supplementary file 4. Statistical results from comparisons of generalized linear models across brain regions. Null generalized linear models (GLMs) and hypothesized (i.e., 'bonding') GLMs were compared with analysis of variance (ANOVA) tests for each region of interest (ROI, 824 regions, $n$ = 189 brain samples, 11-12 samples/group). The ANOVA $F$-statistics are reported alongside p-values computed from permutation tests (10,000 shuffles). The corresponding $q$-values were computed with the false discovery rate (FDR) method to correct for multiple tests.

• MDAR checklist

### Data availability

Source data and source code, along with the prairie vole reference brain and atlas, are available on Figshare (https://doi.org/10.6084/m9.figshare.21375666.v2).

The following dataset was generated:

| Author(s) | Year | Dataset title | Dataset URL | Database and Identifier |
|---|---|---|---|---|
| Gustison M, Muñoz-Castañeda R, Osten P, Phelps S | 2023 | Data and code for "Sexual coordination in a whole-brain map of prairie vole pair bonding | https://doi.org/10.6084/m9.figshare.21375666.v2 | figshare, 10.6084/m9.figshare.21375666.v2 |

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
