## [Editor Report · eLife assessment]

This is an **important** study using 3D mapping of neuronal activation throughout the brain after pair-bonding in the monogamous vole, which can be broadly applied to other species and behaviors. The authors provide **compelling** evidence that there is some synchrony between male and female partners that have formed a pair bond, the strength of which is based on the number of ejaculations received by the female. Same-sex pairs also form a pair bond and were found to have activation in the same brain regions as mixed sex couples. An overall low level of sex differences in the degree and location of brain activation was observed, which was unexpected. This work will be of interest to those interested in social behavior and its neural mechanisms, or brain systems or behavior more broadly.

---

## [Referee Report · Reviewer #1 (Public Review)]

While the approach used in this study cannot identify cause and effect, the whole brain approach identified clusters representing circuits of potential importance and a series of new hypotheses to explore. The importance of the role of sexual behavior, specifically ejaculations rates, is worth emphasizing for the formation of pair bonds. It suggests that the role of sexual behavior in contributing to the strength of pair bonds should be explored more. It is also important to add that males and females in the study were screened for sexual receptivity. The identification of brain regions for pair bond maintenance centered around the amygdala was also intriguing.

---

## [Referee Report · Reviewer #2 (Public Review)]

In this manuscript the authors generate an annotated brain atlas for the prairie vole, which is a widely studied organism. This species has a suite of social behaviors that are difficult or impossible to study in conventional rodents, and has attracted a large community of researchers. The atlas is impressive and will be a fantastic resource. The authors use this atlas to examine brain-wide c-fos expression in prairie voles that were paired with same sex or opposite sex vole across multiple timepoints. In some sense the design resembles PET studies done in primates that take whole brain scans after an important behavioral experience. The authors observed increased c-fos expression across a network of brain regions that largely corresponds with the previous literature. The study design captured several novel observations including that c-fos expression in some regions correlate strongly between males and females during pair bond formation and mating, suggesting synchrony in neural activity. The authors address an important caveat that c-fos provides a snapshot of neural activity and that important populations of neurons could be active and not express c-fos. Thus observed correlations are likely to be robust, but that the absence of differences (in say accumbens) may just reflect the limits of c-fos estimation of neural activity. Similarly, highly coordinated neural activity between males and females might still be driven by different mechanisms if different cell types were activated within a specific region. The creation of this resource and it's use in a well designed study is an important accomplishment.

---

## [Referee Report · Reviewer #3 (Public Review)]

In this manuscript, Gustison et al., describe the development of an automated whole-brain mapping pipeline, including the first 3D histological atlas of the prairie vole, and then use that pipeline to quantify Fos immunohistochemistry as a measure of neural activity during mating and pair bonding in male and female prairie voles. Prairie voles have become a useful animal model for examining the neural bases of social bonding due to their socially monogamous mating strategy. Prior studies have focused on identifying the role of a few neuromodulators (oxytocin, vasopressin, dopamine) acting in limited number of brain regions. The authors use this unbiased approach to determine which areas become activated during mating, cohabitation, and pair bonding in both sexes to identify 68 brain regions clustered in seven brain-wide neuronal circuits that are activated over the course of pair bonding. This is an important study because (i) it generates a valuable tool and analysis pipeline for other investigators in the prairie vole research community and (ii) it highlights potential involvement of many brain regions in regulating sexual behavior, social engagement and pair bonding that have not been previously investigated.

Strengths of the study include the unbiased assessment of neural activity using the automated whole brain activity mapped onto the 3D histological atlas. The design of the behavioral aspect of study is also a strength. Brains were collected at baseline and 2.5, 6 and 22 hrs after cohabitation with either a sibling or opposite sex partner. These times were strategically chosen to correspond to milestones in pair bond development. Behavior was also quantified during epochs over the 22 hr period providing useful information on the progression of behaviors (e.g. mating) during pair bonding and relating Fos activation to specific behaviors (e.g. sex vs bonding). The sibling co-housed group provided an important control, enabling identification of areas specifically activated by sex and bond formation. The analyses of the data were rigorous, resulting in convincing conclusions. While there was nothing particularly surprising in terms of the structures that were identified to be active during the mating and cohabitation, the statistical analysis revealed interesting relationships in terms of interactions of the various clusters, and also some level of synchrony in brain activation between partners. Furthermore, ejaculation was found to be the strongest predictor of Fos activation in both males and females. The sex differences identified in the study was subtle and less than the authors expected, which is interesting.

While the study provides a potentially useful tool and approach that may be general use to the prairie vole community and identifies in an anatomically precise manner areas that may be important for mating or pair bond formation, there are some weaknesses as well. The study is largely descriptive. It is impossible to determine whether the activated areas are simply involved in sex or in the pair bond process itself. In other words, the authors did not use the Fos data to inform functional testing of circuits in pair bonding or mating behaviors. However, that is likely beyond the scope of this paper in which the goal was more to describe the automated, unbiased approach. This weakness is offset by the value of the comprehensive and detailed analysis of the Fos activation data providing temporal and precise anatomical relationships between brain clusters and in relation to behavior. The manuscript concludes with some speculative interpretations of the data, but these speculations may be valuable for guiding future investigations.

---

## [Author Response]

The following is the authors’ response to the original reviews.

**Reviewer #1 (Public Review):**
The importance of the role of sexual behavior, specifically ejaculation rates, is worth emphasizing for the formation of pair bonds in prairie voles. It suggests that the role of sexual behavior in contributing to the strength of pair bonds should be explored more. It is also important to add that males and females in the study were screened for sexual receptivity. It would therefore be important to identify characteristics of animals that did not mate under the laboratory conditions used that may add depth and complexity to what was identified in the current study. The identification of brain regions for pair bond maintenance centered around the amygdala was also intriguing.

Thank you for pointing some interpretations of our findings that can be emphasized in the Discussion. We added the following sentences to the Discussion:

“Our findings, along with this previous work, support the hypothesis that sexual behavior plays a key role in driving pair-bond strength. However, the current study focused on animals that were screened for sexual receptivity, which may have limited variation in sexual behavior across pairs. An intriguing direction for future research will be to test how this variation contributes to bond strength.”

We also emphasized amygdala in relation to pair bond maintenance. We added the following sentence to the Discussion:

“These brain regions, and especially amygdala, will be important candidates for future research on neural regulation of pair-bond maintenance.”

The issue of the lack of a strong presence of the reward circuitry (nucleus accumbens) in the final models is also worth more discussion. Perhaps it has been overly emphasized in the past, but there are strong results from other studies pointing to the importance of reward circuitry.

Thank you for this suggestion. There is a section in the Results that analyses accumbens in more detail than other brain regions. Accumbens did not survive our corrections for multiple statistical tests, however it was significant at early timepoint without these corrections. This Results paragraph states the following:

“Although the nucleus accumbens did not survive multiple test corrections in our ROI analysis (q=0.17), it was significant in univariate analysis (p=0.03), particularly when focused on the 2.5 and 6h timepoints (two sample t-test: t=2.53, p=0.01, Video 2). Furthermore, voxel-level comparisons revealed significant sites within the ventral striatum and the posterior nucleus accumbens (Figure 2A, Figure Supplement 1b-c, Video 2).”

We added Supplementary File 4, which contains model comparison results for accumbens and all other ROIs. We also added more detail on nucleus accumbens to the Discussion:

“Pairing drove increased c-Fos expression in the ventral pallidum, a major node in reward circuity, as well as in the paraventricular nucleus and the medial preoptic area, modulators of reward. This is consistent with a large body of work implicating neuropeptide actions on reward circuits in the formation of bonds (Walum & Young, 2018; Young & Wang, 2004). Conspicuously missing from our list, however, is significant pairing induced c-Fos induction in the nucleus accumbens. One possibility is that an absence of significant accumbens IEG induction reflects the limitations of using c-Fos and other immediate early genes as indicators of neural activity. It is known that some neuronal populations can be active without expressing c-Fos (Sheng & Greenberg, 1990). Indeed, although a variety of studies implicate the accumbens in bond formation (Amadei et al., 2017; Aragona et al., 2006; Scribner et al., 2020), previous work finds only weak c-Fos induction in the prairie vole accumbens during bonding (Curtis & Wang, 2003). Another possibility is that there was heterogeneous activation in the accumbens that was not captured by the precision of our atlas. Consistent with this interpretation, found that the accumbens was significant in univariate tests, as well as in voxel-level analyses. Overall, our results do not conflict with pharmacological, electrophysiological, and calcium-imaging data on the role of the nucleus accumbens in prairie vole bonding (Amadei et al., 2017; Aragona et al., 2006; Scribner et al., 2020). Instead, the absence of significant effects at the level of the entire nucleus accumbens together with the presence of anatomically restricted voxel-level significance suggests substantial anatomical heterogeneity in the contributions of the nucleus accumbens to bond formation.”

Please discuss the consequences of creating the behavioral data for pair bond formation by subtracting same-sex pairs interactions from the opposite-sex interactions. What sources of information are removed by using this approach?

One limitation of our study’s approach is that we are unable to fully separate information related to social novelty from mating experience. Thank you for pointing out that we should touch on this sort of caveat in the paper. We added several sentences to the Discussion:

“It seems likely that sensory and motor areas were important for social processes related to both pair-bonding and reunion with same-sex cagemates, such as investigation and recognition. Our study design, however, highlights differences between treatments, and in order to detect such effects, it might be necessary to compare mating and bonding pairs to animals left in complete isolation.”

We reiterate the point in a new paragraph we added to the Discussion to explicitly provide caveats regarding our data:

“Before offering a synthesis of our findings, it would be useful to acknowledge a few caveats. First, as noted above, IEG induction does not capture all relevant neural activity. Second, the design of our experiment, which controlled for social interaction, likely excluded many circuits important to both pair bonding and sibling social interactions. Third, c-Fos activity within a given brain region may nevertheless rely on distinct cell types, and so the absence of sex differences in c-Fos immunoreactivity does not definitively rule out the sexually dimorphic circuits hypothesized in the “dual function hypothesis” (de Vries, 2004). Lastly, the current study focused on animals that were screened for sexual receptivity, which may have limited the variation in sexual behavior across opposite-sex pairs.”

Time 0 is when the barrier is removed after a two-hour exposure. Please speculate on what is going on during the two-hour exposure. Time zero is potentially more than the time of mating. Is it possible that aggression is being decreased during this timepoint that represents mating? Could it also be a measure of the outcome of an initial compatibility assessment by the male and female?

Thank you for this interesting observation. While the opaque divider prevented physical social interactions, it is possible that animals picked up on auditory or olfactory cues. We did not detect group differences in movement patterns and vocalization rates from the 0 h timepoint group (Figure 2). These findings suggest that potential partner detection and assessment occurred in a similar way for both experiment groups. It is unlikely that this period represents a decrease in aggression, since unbonded prairie voles are not known to be aggressive towards conspecifics. However, the idea that animals may potentially use olfactory or auditory cues to assess each other is an interesting idea, and one that we cannot rule out. We added a brief statement to the Methods “Experiment Design” section about the possibility that the two hours prior to divider removal (0 h timepoint) could represent more than an acclimation period:

“It is important to note that the opaque divider in the acclimation period prevented physical interactions, but it is possible that animal pairs may have detected each other through olfactory or auditory cues.”

We also mention this in the revised Discussion in the context of the PFC cluster, which not only differed between mating and non-mating groups, but also showed differences between isolated (0h) and socially interacting animals (sibs and mates, 2.5h-22h):

“A fourth cluster (“PFC,” green) is composed of prelimbic, infralimbic and olfactory cortex; activity in the vole prefrontal cortex is known to be modulated by hypothalamic oxytocin, and to shape bonding through projections to the nucleus accumbens (Amadei et al., 2017; Burkett et al., 2016; Horie et al., 2020). The pattern of activity in this cluster, however, indicates that it was due in part to differences between the isolated animals (0h) and other time points (Figure 4—figure supplement 1 and Figure 4—figure supplement 2). Because animals in the isolated condition were in a compartment adjacent to either an opposite sexed individual or a familiar former cagemate, we cannot rule out that olfactory or auditory cues may have made animals aware of the presence of a potential social partner. Indeed, we interpret this dimension as capturing appetitive aspects of behaviors associated with investigation of the animal isolated from the subject by the barrier.”

**Reviewer #2 (Public Review):**
An important caveat to this study not mentioned by the authors is that c-fos provides a snapshot of neural activity and that important populations of neurons could be active and not express c-fos. Thus observed correlations are likely to be robust, but the absence of differences (in say accumbens) may just reflect the limits of c-fos estimation of neural activity. Similarly, highly coordinated neural activity between males and females might still be driven by different mechanisms if different cell types were activated within a specific region.

We now discuss limitations of c-Fos in the Discussion paragraph that focuses on accumbens:

“The absence of significant accumbens IEG induction may reflect the limitations of using c-Fos and other immediate early genes as indicators of neural activity. It is known that some neuronal populations can be active without expressing c-Fos (Sheng & Greenberg, 1990). Indeed, although a variety of studies implicate the accumbens in bond formation (Amadei et al., 2017; Aragona et al., 2006; Scribner et al., 2020), previous work finds only weak c-Fos induction in the prairie vole accumbens during bonding (Curtis & Wang, 2003).”

We also include the following sentence in a new Discussion paragraph that focuses on caveats to our findings:

“Before offering a synthesis of our findings, it would be useful to acknowledge or reiterate a few caveats. First, as noted above, IEG induction does not capture all relevant neural activity (Sheng & Greenberg 1990). Second, the design of our experiment, which controlled for social interaction, likely excluded many circuits important to both pair bonding and sibling social interactions. Third, c-Fos activity within a given brain region may nevertheless rely on distinct cell types, and so the absence of sex differences in c-Fos immunoreactivity does not definitively rule out the sexually dimorphic circuits hypothesized in the “dual function hypothesis” (de Vries, 2004). Lastly, the current study focused on animals that were screened for sexual receptivity, which may have limited the variation in sexual behavior across opposite-sex pairs.”

**Recommendations for the authors:**
It appears as if df is missing from some statistical reporting.

Thank you for pointing this out. We went through the manuscript and added in sample sizes to statistical reporting.

**Reviewer #1 (Recommendations for the authors):**
It is surprising that the cortex was not more extensively identified as being involved in pair bonding, but perhaps this is because the emphasis for choosing brain areas in the cortical region is biased towards olfactory regions. Please discuss. It may also be worth noting that brain regions associated with perception may be important in all of these processes, but selected out because of the design.

Thank you for this observation. We agree that some cortical regions may not have been identified due to the study design. For example, social processes related to both pair bonding and cagemate recognition likely rely on overlapping circuits. It is also important to note here that our analysis approach identified the “most” significant regions. This means that several candidate regions did not survive the statistical threshold used to select regions. We now discuss the cortex in more detail in the Discussion, where we also identify the regions that approached significance but did not survive multiple test corrections:

“Although the PFC and other olfactory cortical areas formed a cluster, we did not find widespread c-Fos induction throughout the cortex in response to pairing. It seems likely that sensory and motor areas were important for social processes related to both pair-bonding and reunion with same-sex cagemates, such as investigation and recognition. Our study design, however, highlights differences between treatments, and in order to detect such effects, it might be necessary to compare mating and bonding pairs to animals left in complete isolation. Moreover, several cortical regions that did not survive corrections for multiple tests may have been identified in a less stringent analysis. Several subregions within the isocortex, hippocampal formation, and cortical subplate had statistical models that approached significance (i.e., p-values < 0.1) prior to multiple test corrections. These subregions were found within primary somatosensory area, primary auditory area, dorsal and ventral auditory areas, primary visual area, anteromedial visual area, agranular insular area, temporal association areas, ectorhinal area, postsubiculum, and basomedial amygdala. Frontal cortex subregions were within the agranular insular area and orbital area, as well as additional subregions in prelimbic and infralimbic areas of the PFC.”

Same-sex siblings were isolated for 4-5 days and then repaired. This is a creative way of dealing with this, but was any aggression displayed in the same-sex pairs? Are there bonds or preferences among same-sex individuals? Could the isolation have set the stage for neural changes associated with migrating from the natal group? 4-5 days of isolation is not trivial.

Thank you for these questions. We did not witness aggression between same-sex pairs. We had recorded ‘aggression’ events (lunges and chases) during the 1 h behavioral observation epochs and found that these rates were nearly zero for all sibling timepoint groups (events/h per focal animal in mean ± sd: 2.5 h group = 0.58 ± 1.53, 6 h group = 0.17 ± 0.48, 22 h group = 0.25 ± 0.44).

The question about peer relationships is a good one. Previous literature does suggest that prairie voles can develop preferences for familiar same-sex individuals (e.g., Beery et al. 2018 Front. Behav. Neuro., Lee et al. 2019 Front. Behav. Neuro). Thus, we want to reiterate here that our study design tests for differences between these baseline levels of affiliation with pair bonding in a reproductive context.

It is possible that the period of isolation prior to experiments may have set the stage for neural changes associated with migration from the natal group. Testing this possibility is outside the scope of the current study. We want to point out here that animals were separated from their natal groups several weeks prior to the experiment. Animals were weaned at 21 days and put into same-sex cages, and then experiments occurred between 8-12 weeks of age. All experiment groups went through the same weaning and co-housing conditions.

Pg 26, Line 655: "better" is listed twice in the sentence and only one is needed

Thank you for catching this typo. This is fixed.

**Reviewer #2 (Recommendations for the authors):**
Why was it necessary to bring voles into estrus when they are induced ovulators? The authors need to state how voles were brought into estrus.

Thank you for this suggestion. We explained estrus induction in the Methods, but this explanation could be missed because it was within the “Behavioral procedures” section. We put the paragraph about estrus induction into a new section called “Estrus induction and animal selection”. We also elaborated on the final sentences of this paragraph to provide a clearer rationale:

“We used this mating assay to restrict study subjects to voles that showed lordosis (females) or mounting behavior (males). By selecting voles who showed sexual behavior, we could control the estrus state and timing of mating across the 0, 2.5, 6 and 22 h study groups. This selection process also ensured that animals assigned to the same-sex sibling pair and opposite-sex mating pair groups had similar sexual motivation and experience.”

I assume in the final manuscript the authors will release the availability of the atlas? Making the atlas public seems to be in the spirit of the eLife publishing model.

The prairie vole reference brain, atlas, and atlas annotation labels, are now included on the Figshare repository site. We updated the Data and code availability section to clarify this.

**Reviewer #3 (Recommendations for the authors):**
Please clarify in the Methods if same-sex sibling females were also estrogen primed. If not, could the estrogen exposure cause Fos differences?

Thank you for this suggestion. All females were estrogen primed. We refined the Methods section “Estrus induction and animal selection” to make this part of the study design clearer. We edited one of the sentences to say “During this isolation period, all females were induced into estrus[...]” We also added a couple sentences at the end of this paragraph:

“By selecting voles who showed sexual behavior, we could control the estrus state and timing of mating across the 0, 2.5, 6 and 22 h study groups. This selection process also ensured that animals assigned to the same-sex sibling pair and opposite-sex mating pair groups had similar sexual motivation and experience.”